# Structural basis of the activation of TRPV5 channels by long-chain acyl-Coenzyme-A

Bo-Hyun Lee [1,2,4], José J. De Jesús Pérez[3,4], Vera Moiseenkova-Bell [3] ✉ & Tibor Rohács [1] ✉

Long-chain acyl-coenzyme A (LC-CoA) is a crucial metabolic intermediate that plays important cellular regulatory roles, including activation and inhibition of ion channels. The structural basis of ion channel regulation by LC-CoA is not known. Transient receptor potential vanilloid 5 and 6 (TRPV5 and TRPV6) are epithelial calcium-selective ion channels. Here, we demonstrate that LC-CoA activates TRPV5 and TRPV6 in inside-out patches, and both exogenously supplied and endogenously produced LC-CoA can substitute for the natural ligand phosphatidylinositol 4,5-bisphosphate ($PI(4,5)P_2$) in maintaining channel activity in intact cells. Utilizing cryo-electron microscopy, we determined the structure of LC-CoA-bound TRPV5, revealing an open configuration with LC-CoA occupying the same binding site as $PI(4,5)P_2$ in previous studies. This is consistent with our finding that $PI(4,5)P_2$ could not further activate the channels in the presence of LC-CoA. Our data provide molecular insights into ion channel regulation by a metabolic signaling molecule.

LC-CoA esters are metabolic intermediates that are involved in the synthesis of fatty acids. They have been shown to play a role in regulating various cellular processes, including mitochondrial metabolism, insulin secretion and signaling, and gene transcription[1]. LC-CoA has been shown to modulate the activity of various ion channels, the best characterized of which are ATP sensitive $K^+$-channels ($K_{ATP}$), which act as metabolic sensors in a variety of tissues including pancreatic β-cells and cardiac myocytes[2–4]. It has been proposed that LC-CoA accumulation in pancreatic β-cells in type 2 diabetes promotes $K_{ATP}$ channel opening and reduces glucose sensitivity of insulin secretion[2]. Similarly, in cardiomyocytes, it has been suggested that increased LC-CoA levels during ischemia contribute to opening of $K_{ATP}$ channel thus decreasing cardiac activity[5]. Although it has been suggested that LC-CoA activates $K_{ATP}$ channels in a similar manner to the endogenous cofactor $PI(4,5)P_2$[3,4], structural data supporting such mechanism is currently lacking.

LC-CoA also activates the heat- and capsaicin-sensitive Transient Receptor Potential Vanilloid 1 (TRPV1) channel[6,7]. While its phosphoinositide regulation is complex[8], $PI(4,5)P_2$ reliably activates TRPV1

channels in excised inside-out patches[6,9,10]. Similar to $K_{ATP}$ channels[3], TRPV1 exhibits low specificity for $PI(4,5)P_2$ and can be activated by a number of negatively charged lipids, including LC-CoA[6]. However, there is a lack of direct evidence for the mechanism by which LC-CoA regulates ion channels, as no ion channel structures with LC-CoA have been experimentally determined yet.

TRPV5 and TRPV6 are $Ca^{2+}$ selective channels that play key roles in transcellular $Ca^{2+}$ transport in epithelial cells, including those in kidneys, intestines and the exocrine pancreas[11]. TRPV5 plays a crucial role in $Ca^{2+}$ reabsorption in the distal tubule, and its mutations are associated with kidney stones[12]. TRPV6 was originally identified in the duodenum, where it plays a role in $Ca^{2+}$ absorption, but it is also expressed in various other epithelial cell types including those in the exocrine pancreas. Loss of function mutations in TRPV6 are associated with pancreatitis (heterozygous)[13] and with Neonatal Hyperparathyroidism (homozygous)[14]. Both TRPV5 and TRPV6 are constitutively active and require the presence of the membrane phospholipid $PI(4,5)P_2$ for this activity[15–17]. The structure of TRPV5 has been determined earlier with $PI(4,5)P_2$ showing an open

[1]Department of Pharmacology, Physiology and Neuroscience, Rutgers, New Jersey Medical School, Newark, NJ, USA. [2]Department of Physiology, Gyeongsang National University Medical School, Jinju, Korea. [3]Department of Systems Pharmacology and Translational Therapeutics, University of Pennsylvania, Philadelphia, PA, USA. [4]These authors contributed equally: Bo-Hyun Lee, José J. De Jesús Pérez. ✉e-mail: vmb@pennmedicine.upenn.edu; tibor.rohacs@rutgers.edu

structure, while in the absence of this lipid the protein was in a closed conformation[18,19].

In this study, we identify TRPV5 and TRPV6 as novel LC-CoA-activated ion channels. Our functional studies indicate that the activation of TRPV5 and TRPV6 channels by LC-CoA required both a long acyl chain and the 3' phosphate in its head group. In agreement with these results, our cryoEM structure of TRPV5 in the absence of LC-CoA remained closed, while in complex with LC-CoA it showed an open state, not only mimicking the conformational state determined earlier with PI(4,5)P$_2$, but also conserving the activating lipid-binding site. We also show that both exogenously supplied and endogenously produced LC-CoA can substitute for PI(4,5)P$_2$ in maintaining channel activity in a cellular context. Our data provide definitive evidence for opening of an ion channel by LC-CoA by binding to its phosphoinositide binding site, and highlights the importance of this shared binding site in the opening of TRPV5 channels.

## Results

### Activation of TRPV5 and TRPV6 by CoA species requires a long acyl chain

We studied the effect of long-chain acyl-CoA (LC-CoA) in excised inside-out patches on the rabbit TRPV5 and the human TRPV6 channels expressed in *Xenopus laevis* oocytes. These channels are known to require PI(4,5)P$_2$ for activity[15,17]. Since LC-CoA and PI(4,5)P$_2$ have structural similarities such as three phosphate groups in the head group and one or two acyl chains (Fig. 1a, b), we hypothesized that LC-CoA can substitute for PI(4,5)P$_2$ in supporting TRPV5 and TRPV6 activity. To test this hypothesis, we examined the effect of different types of fatty acyl-CoA on both TRPV5 and TRPV6 (Fig. 1). As a control, water-soluble diC$_8$ PI(4,5)P$_2$ was used to reversibly activate the channels. Table 1 shows six different fatty acyl-CoA species used in the current study. Their names are represented as (chain length: number of the double bonds), for instance octanoyl-CoA is denoted as C8:0 CoA.

After establishing the inside-out configuration, TRPV5 and TRPV6 show current run-down, which is due to dephosphorylation of endogenous PI(4,5)P$_2$[20]. After run-down, the application of 25 μM diC$_8$ PI(4,5)P$_2$ induced currents for both TRPV5 (Fig. 1c) and TRPV6 (Fig. 1e). DiC$_8$ PI(4,5)P$_2$ (Fig. 1b) is a short acyl chain synthetic analog of PI(4,5)P$_2$ which is often used as an experimental tool, because it activates ion channels, including TRPV5 and TRPV6 quickly and reversibly, as opposed to the slow and almost irreversible effect of natural long acyl chain PI(4,5)P$_2$[20,21].

Oleoyl-CoA (C18:1) reactivated both TRPV5 and TRPV6 in a concentration-dependent manner (Fig. 1c–f). TRPV5 currents induced by 10 μM C18:1 CoA showed 6.19 ± 2.94 fold higher currents than those induced by 25 μM diC$_8$ PI(4,5)P$_2$ (Fig. 1c, d), which is a sub-maximal concentration for channel activation[22]. The shorter acyl chain octanoyl CoA (C8:0) showed no effect even at 100 μM. Similar results were obtained with TRPV6; no currents were induced by the application of C8:0 CoA up to 100 μM, but C18:1 CoA activated TRPV6 in a concentration-dependent manner (Fig. 1e, f). TRPV6 currents induced by 10 μM C18:1 CoA were 4.3 ± 0.86 fold higher than those induced by 25 μM diC$_8$ PI(4,5)P$_2$ (Fig. 1e, f). Note that application of C18:1 CoA and other LC-CoA species made the patches unstable, thus we avoided long applications, and for this reason the currents usually did not reach steady state. Also, the concentration response relationship for both channels was very steep, with no, or only marginal effect at 1 μM and maximal effect at 10 μM, similar to that observed in K$_{ATP}$ channels[3].

Our data so far suggest that a long acyl chain is required for channel activation by CoA. Next, we performed a more detailed characterization of the acyl-chain length dependence. Acyl-CoA species with increasing chain lengths (C8:0, C12:0, C14:0, C16:0, C18:0, and C18:1) were applied at 10 μM to induce channel activation (Fig. 1g, h, and Supplementary Fig. 1). Both channels were activated starting at

C12:0 and the currents were getting larger up to C16:0. However, C18:0 and C18:1 CoA did not exert any further increase in currents indicating 10 μM C16:0 CoA may already cause full activation of the channel. Stearyl- (C18:0) and oleoyl- (C18:1) CoA induced comparable activation of both TRPV5 (Fig. 1g) and TRPV6 (Fig. 1h). These data suggest that the efficiency of fatty acyl-CoA to activate TRPV5 and TRPV6 correlated with increasing acyl chain length, and the number of double bonds in the fatty acid tail did not have a major influence on its effectiveness.

Oleoyl-CoA (10 μM) did not induce any currents in excised inside-out patches from non-injected oocytes (Supplemental Fig. 2), indicating lack of LC-CoA-activated endogenous currents in oocytes not expressing TRPV5 or TRPV6.

### The 3'-phosphate is required for full activation by LC-CoA

To test the role of the number of phosphate groups, we applied 3'-dephospho-palmitoyl-CoA, which lacks the 3'-phosphate (Fig. 2a) in excised inside out patches. Figure 2b, d show that 10 μM 3'-dephospho-palmitoyl-CoA had little or no effect on the activity of TRPV5 and TRPV6 compared to C16:0 CoA. The application of 10 μM C16:0 CoA produced robust activation in the same patches, 8.35 ± 1.93-fold increase compared to the current induced by 25 μM diC$_8$ PI(4,5)P$_2$, while the same concentration of 3'-dephospho-palmitoyl-CoA produced no, or weaker activation (0.31 ± 0.29-fold) on TRPV5 (Fig. 2c). Similarly, TRPV6 was strongly activated by 10 μM C16:0 CoA (12.00 ± 1.41-fold), and only to a much smaller extent by 10 μM 3'-dephospho-palmitoyl-CoA (0.54 ± 0.20-fold) (Fig. 2e). Our finding is consistent with the previous report that 3'-dephospho-palmitoyl-CoA does not activate K$_{ATP}$ channels[23]. Taken together, these results indicate that the presence of the 3'-phosphate in the CoA moiety plays a role in TRPV5 and TRPV6 activation. This dependence of activity on the presence of three phosphates is similar to the lack of activation of TRPV6 by PI(4)P[17].

The effect of LC-CoA was reversible in all experiments, but the return to baseline was slow and variable after washout of the lipid (Fig. 1). To accelerate deactivation, in some experiments we applied bovine serum albumin (BSA, 5 mg/ml) which binds lipids after LC-CoA application to extract LC-CoA from the membrane, which greatly accelerated the deactivation of the channel (Figs. 2b and 3b, d).

Next, we tested the effects of the synthetic lipid, DGS-NTA, which carries two C18:1 lipid tails, and three negatively charged carboxyl groups (Fig. 3a). DGS-NTA was shown to activate both K$_{ATP}$[24] and TRPV1[6] channels. Representative traces show that 10 and 100 μM DGS-NTA did not activate TRPV5 (Fig. 3b) and TRPV6 (Fig. 3d) in excised patches at −100 mV. Figure 3c, e shows summary of the effect of DGS-NTA compared to 25 μM diC$_8$ PI(4,5)P$_2$ and 10 μM C18:1 CoA on TRPV5 and TRPV6. These data show that activation by LC-CoA is specific to this lipid, and not a non-specific activation by lipids with long acyl chain and negatively charged headgroup.

### CryoEM reveals that LC-CoA opens TRPV5 by binding to the same site as PI(4,5)P$_2$

To understand the molecular mechanism of the activation of TRPV5 by LC-CoA, we solved the cryoEM structure of the full-length rabbit TRPV5 in complex with oleoyl-CoA (Fig. 4, Supplementary Figs. 3−5, Table 2). The purified TRPV5 in lipid nanodiscs was incubated with 400 μM oleoyl-CoA for 45 min before freezing, and processing of the dataset revealed two different states: The TRPV5$_{CoA\ Open}$ at a resolution of 3.3 Å (Fig. 4a) and the TRPV5$_{CoA\ Closed}$ at 3.1 Å (Fig. 4c). The TRPV5$_{CoA\ Open}$ structure shows a clear elongated density between the S2-S3 linker, S3, S4-S5 linker, and S6 helix, which is consistent with an oleoyl-CoA molecule with the adenosine 3',5'-diphosphate at the cytosolic side (Fig. 4b). On the other hand, the TRPV5$_{CoA\ Closed}$ structure does not have such additional density (Fig. 4d). The CoA-binding site is constituted by the Q483, K484 and R492 (S4-S5 linker), and the R584 (S6 helix) (Fig. 4b), the same residues that were reported for

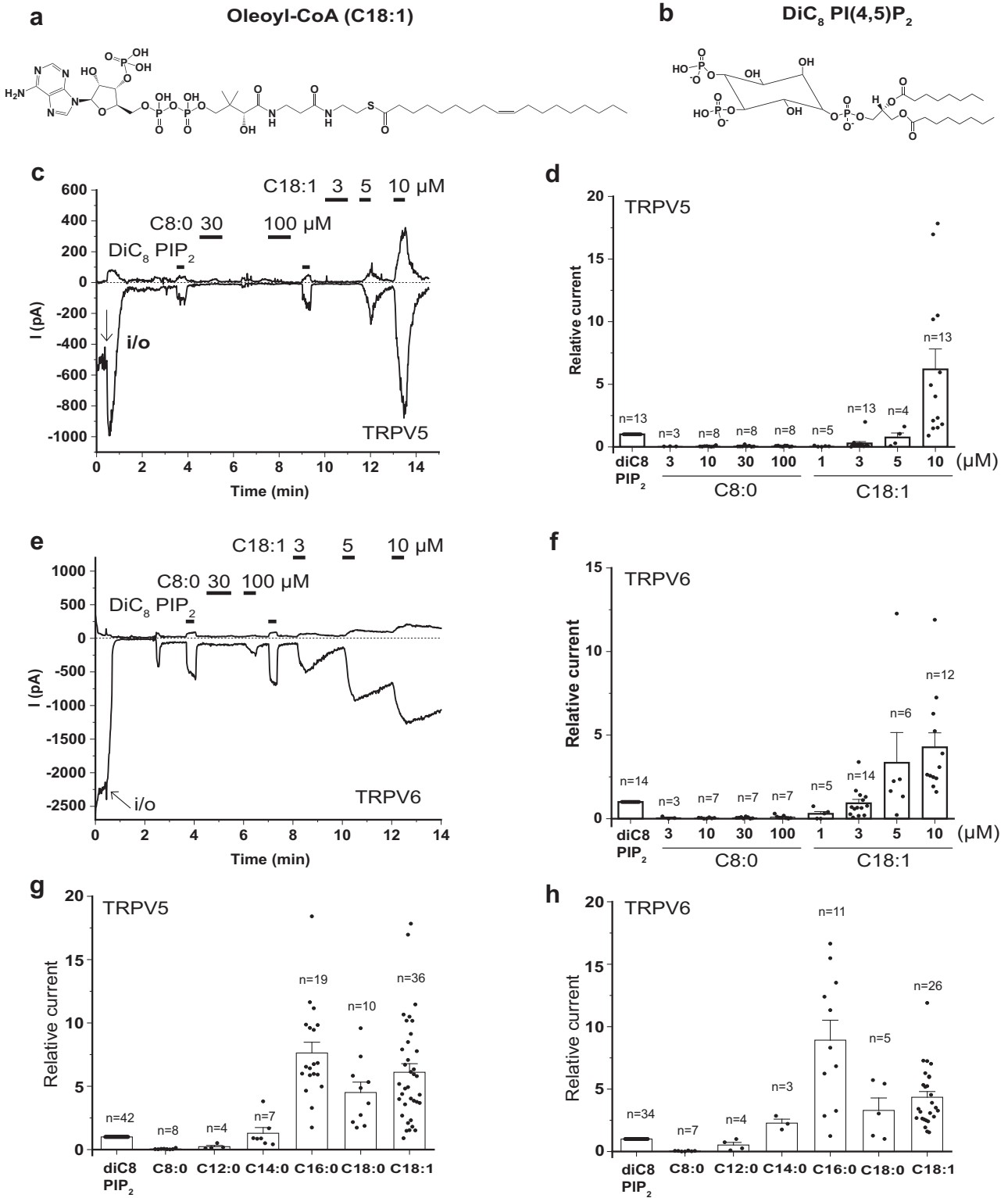

**Fig. 1 | Long chain acyl-CoA activates TRPV5 and TRPV6 channels.** Chemical structures of oleoyl-CoA (C18:1) (**a**) and dioctanoyl (diC$_8$) PI(4,5)P$_2$ (**b**). **c–h** Excised inside-out patch clamp recordings on TRPV5- and TRPV6-expressing *Xenopus laevis* oocytes were performed as described in the Methods. Traces show currents at +100 (upper) and −100 (lower) mV. Dashed lines show zero current. The establishment of the inside-out (i/o) configuration is indicated by the arrows. The applications of 25 μM diC$_8$ PI(4,5)P$_2$ (PIP$_2$) and different concentrations of octanoyl (C8:0) CoA, and C18:1 CoA are shown by the horizontal lines. Representative trace for the effects of C8:0 CoA and C18:1 CoA on TRPV5 (**c**) and TRPCV6 (**e**). Summary of the effects of C8:0 CoA and C18:1 CoA on TRPV5 (**d**) and TRPV6 (**f**). Summary of the effects of the different fatty acyl-CoAs (10 μM) on TRPV5 (**g**) and TRPV6 (**h**). Bar graphs show mean ± SEM and scatter plots. The symbols represent individual oocytes.

## Table 1 | Fatty acyl-coenzyme A (CoA) species used in this study

| Fatty acyl-CoA | Chain length | Double bond |
|---|---|---|
| Octanoyl-CoA | C8 | 0 |
| Lauroyl-CoA | C12 | 0 |
| Myristoyl-CoA | C14 | 0 |
| Palmitoyl-CoA | C16 | 0 |
| Stearoyl-CoA | C18 | 0 |
| Oleoyl-CoA | C18 | 1 |

the PI(4,5)P$_2$-binding site[18,19] (Fig. 4i). Compatible with this result, the TRPV5$_{CoA\ Open}$ structure resembles the open state of TRPV5 in the presence of PI(4,5)P$_2$ (TRPV5$_{PIP2}$) with a RMSD = 0.8 Å (Fig. 4e, f, h, k), and TRPV5$_{CoA\ Closed}$ resembled the TRPV5 apo state (TRPV5$_{Apo}$) with a RMSD = 0.5 Å (Fig. 4g, h). Similar to the way PI(4,5)P$_2$ activates TRPV5, the third phosphate of CoA induce the rotation of the R584 on S6 out of the pore by 39°, forming a salt bridge with this residue (Fig. 4l). Parallel to this, W583 turns out by 129° releasing the pore entrance (Fig. 4l). These conformational changes pull the S6 helix away from the pore, causing the opening of the lower gate and the activation of the channel (Fig. 4e–h). Our results indicate that LC-CoA and PI(4,5)P$_2$ activate TRPV5 channels through the same binding site.

### PI(4,5)P$_2$ does not further activate TRPV5 and TRPV6 after LC-CoA activation

The cryoEM structure shows that PI(4,5)P$_2$ and LC-CoA activate TRPV5 by binding to the same binding site. This predicts that in the presence of LC-CoA, PI(4,5)P$_2$ would not further activate the channels. To test this prediction, we co-applied C18:1 CoA and diC$_8$ PI(4,5)P$_2$ in excised inside-out patches (Fig. 5). First, currents induced by 3 μM or 10 μM C18:1 CoA (area 1, light gray in Fig. 5a, b) were recorded, then a mixture of C18:1 CoA and 25 μM diC$_8$ PI(4,5)P$_2$ (area 2, dark gray in Fig. 5a, b) was applied then a C18:1 CoA only solution (area 3, light gray in Fig. 5a, b). To better understand the effect of C18:1 CoA, the average currents from area 1 and 3 were calculated and compared to area 2. When the channel was activated by 3 μM C18:1 CoA, 25 μM diC$_8$ PI(4,5)P$_2$ did not induce further activation of TRPV5 (Fig. 5c) and TRPV6 (Fig. 5e). When the channels were activated by 10 μM C18:1 CoA, 25 μM diC$_8$ PI(4,5)P$_2$ also did not induce further TRPV5 activation (Fig. 5d) but it slightly, but statistically significantly reduced TRPV6 currents (p = 0.013) (Fig. 5f). A similar trend without reaching statistical significance was also observed on TRPV5 with 10 μM C18:1 CoA (Figs. 5d) and 3 μM C18:1 CoA with TRPV6 (Fig. 5e). These observations support the idea that C18:1 CoA act on same binding site as PI(4,5)P$_2$, and suggest that diC$_8$ PI(4,5)P$_2$ is a somewhat less efficacious channel activator than LC-CoA.

### Exogenous LC-CoA can substitute for endogenous PI(4,5)P$_2$ in maintaining channel activity in intact cells

Our excised patch data demonstrate that LC-CoA can substitute for PI(4,5)P$_2$ in activating TRPV5 and TRPV6 channels. To test if LC-CoA can exert similar effects in a cellular environment, we performed whole cell patch clamp experiments, where we applied LC-CoA, or diC$_8$ PI(4,5)P$_2$ through the patch pipette. To deplete endogenous PI(4,5)P$_2$, we used the *ciona intestinalis* voltage-sensitive 5' phosphatase (ci-VSP), which is a membrane protein that converts PI(4,5)P$_2$ to PI(4)P, a lipid that was shown not activate TRPV6[17]. We used HEK293 cells co-transfected with TRPV6 and ci-VSP for whole-cell voltage clamp experiments. Monovalent TRPV6 currents were measured at a holding potential of −60 mV, and once every minute ci-VSP was activated by depolarizing pulses to +100 mV for 1 s and this pulse step was repeated 14 times (Fig. 6a). The voltage steps induced a fast and robust inhibition of channel activity (79.7 ± 1.9% inhibition on average across pulses 1–9), followed by a slower recovery, until the next voltage pulse was

applied. This recovery is due to resynthesis of PI(4,5)P$_2$ by PI(4)P 5-kinase enzymes[25]. Since the intracellular pipette solution did not contain ATP, this recovery was partial and currents gradually decreased throughout the experiment (Fig. 6a), presumably because of the loss of cellular ATP. In control cells, TRPV6 currents were gradually reduced when VSP hydrolyzed endogenous PI(4,5)P$_2$ resulting in a reduction of the current to 73 ± 8% of the initial current on average after the 3$^{rd}$ pulse and to 31 ± 9% after the 9$^{th}$ pulse (Fig. 6a, d).

Next, we tested whether TRPV6 current activity can be recovered by supplementing the intracellular pipette solution with 100 μM diC$_8$ PI(4,5)P$_2$ (Fig. 6b), or 10 μM C16:0 CoA (Fig. 6c). When we used 100 μM diC$_8$ PI(4,5)P$_2$, relative currents were maintained throughout the experiment, due to the continuous replenishment of diC$_8$ PI(4,5)P$_2$ from the intracellular pipette solution (99 ± 3%) (Fig. 6b, d). However, depolarizing pulses acutely inhibited the currents to a similar extent to control cells (81.6 ± 1.5% inhibition) (Fig. 6b, e), indicating that ci-VSP is capable of hydrolyzing diC$_8$ PI(4,5)P$_2$ in the plasma membrane, but the excess lipid from the patch pipettes recovers currents between depolarizing pulses.

Supplementing with 10 μM C16:0 CoA also showed maintained relative currents similar diC$_8$ PI(4,5)P$_2$ (94.2 ± 3.5%) by the end of the experiment (Fig. 6c, d), indicating LC-CoA is capable of maintaining TRPV6 activity. After the depolarization pulses, only minimal acute inhibition was measured (21 ± 2% inhibition on average) (Fig. 6d, e), likely due to hydrolysis of remaining endogenous PI(4,5)P$_2$ with most of channel activity maintained by LC-CoA, which is not hydrolyzed by the phosphatase. These finding demonstrate that LC-CoA (C16:0) is capable of supporting TRPV6 activity in a cellular environment.

### Increasing endogenous LC-CoA levels reduces the dependence of TRPV6 activity on PI(4,5)P$_2$

Next, we tested whether increased endogenous LC-CoA levels can also substitute for PI(4,5)P$_2$ and maintain channel activity. We used a protocol similar to that previously shown to increase cellular LC-CoA levels, and stimulate the activity of K$_{ATP}$ channels[23]. In these experiments HEK293 cells expressing TRPV6 and ci-VSP were treated overnight with 164 μM oleic acid in complex with albumin in a 2:1 molar ratio, in the presence of high glucose (25 mM). High glucose is known to promote accumulation of LC-CoA in the cytoplasm, via formation of high concentrations of malonyl-CoA, which in turn inhibits the carnitine-palmitoyl-transferase-1 (CPT1), thus reducing transport of LC-CoA to the mitochondria (Fig. 7f). We found that cells treated with oleic acid in the presence high glucose had reduced inhibition by ci-VSP compared to cells without treatment with oleic acid (Fig. 7a, b, e).

Cells treated with oleic acid in the presence of low glucose (5.5 mM) on the other hand were inhibited by ci-VSP activation to a similar level to control cells (Fig. 7c, e). When cells kept in normal glucose and oleic acid were treated cells with the CPT1 inhibitor etoxomir, they also showed reduced inhibition by ci-VSP (Fig. 7d, e). These data indicate that increased cytoplasmic LC-CoA levels can reach the plasma membrane in high enough concentrations to reduce dependence of channel activity on PI(4,5)P$_2$.

## Discussion

LC-CoA is a metabolic intermediate that plays important cellular regulatory roles, including activating and inhibiting ion channels. Currently, there is no structural information on how this lipid binds to and modulates ion channels. Here we identified LC-CoA as a highly efficient activator of TRPV5 and TRPV6 channels. We find that both the presence of a long acyl chain, and the presence of the 3' phosphate in LC-CoA are important for activating these channels. We determined the cryoEM structure of TRPV5 in the presence of oleoyl-CoA, and show that this lipid binds to the same site as the endogenous ligand PI(4,5)P$_2$ and induces very similar conformation changes to open the channel. Our work provides structural insight into how LC-CoA opens an ion

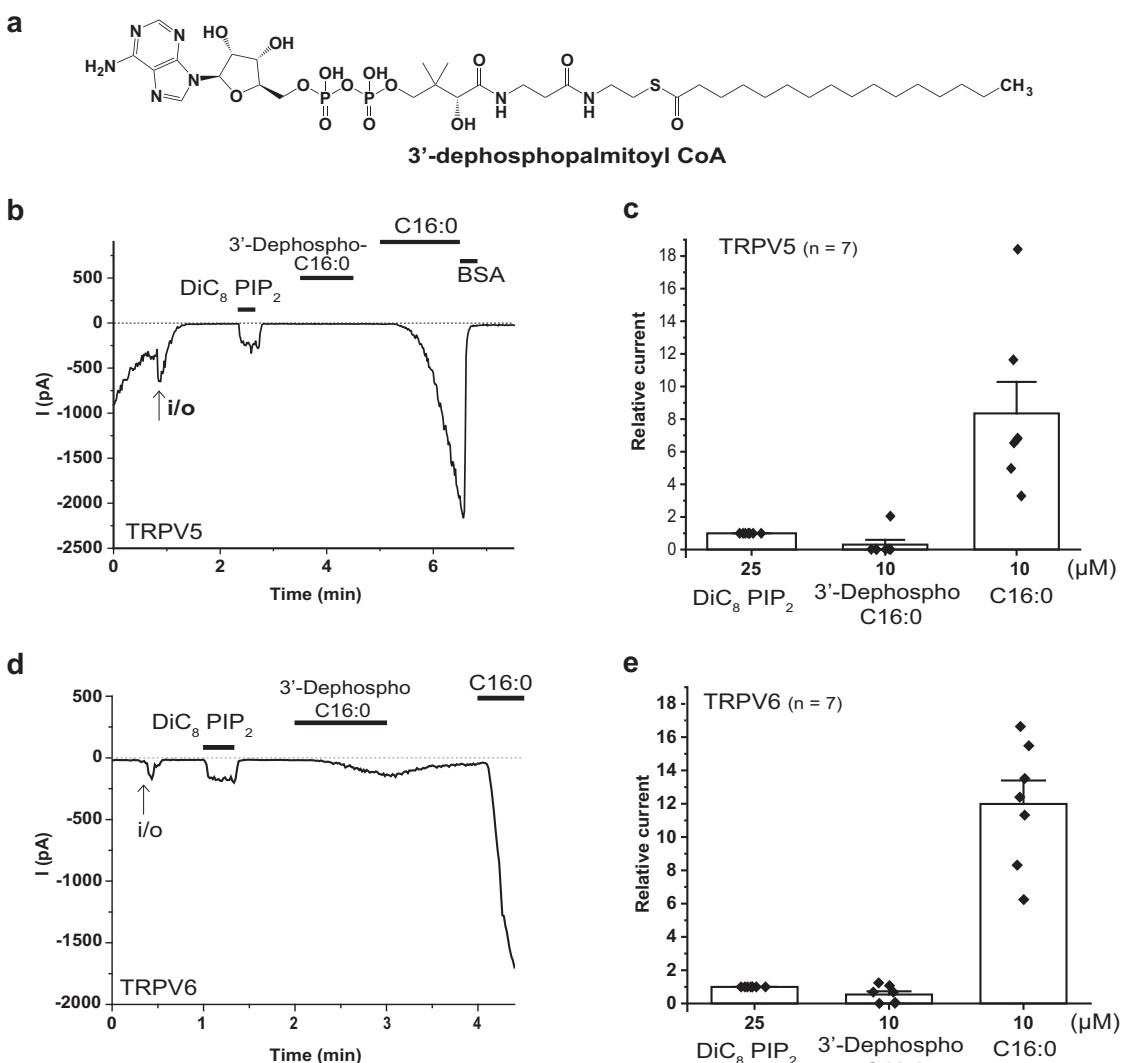

**Fig. 2 | The 3'-phosphate is required for LC-CoA to activate TRPV5 and TRPV6 channels. a** Chemical structure of 3'-dephospho-palmitoyl CoA. **b-e** Excised inside-out patch clamp recordings on *Xenopus laevis* oocytes expressing TRPV5 and TRPV6 were performed as described in Methods. Traces show currents at −100 mV, dashed lines show zero current. The establishment of the inside-out configuration (i/o) is indicated by the arrows. The application of 25 μM diC$_8$ PI(4,5)P$_2$, 10 μM 3'- dephospho-palmitoyl-CoA, and 10 μM palmitoyl (C16:0) CoA are shown by the horizontal lines. Representative traces for the effect of diC$_8$ PI(4,5)P$_2$, 3'-depho-spho-palmitoyl-CoA, and C16:0 CoA on TRPV5 (**b**) and TRPV6 (**d**). Summary of the effects of diC$_8$ PI(4,5)P$_2$, 3'-dephospho-palmitoyl-CoA, and C16:0 CoA on TRPV5 (**c**) and TRPV6 (**e**). Bar graphs show mean ± SEM and scatter plots. The symbols represent experiments from individual oocytes.

channel, and it also points to central role of the PI(4,5)P$_2$ binding site in TRPV5 gating.

Among all LC-CoA-regulated ion channels, K$_{ATP}$ channels are the best characterized. K$_{ATP}$ channels consist of a pore forming inwardly rectifying K$^+$ channel subunit Kir6.2 or Kir6.1 and a sulfonylurea subunit SUR1 or SUR2B. Most other members of the Kir channel family are inhibited by LC-CoA[3,26]. All Kir channels require the membrane phospholipid PI(4,5)P$_2$ for activity, and they show variable selectivity between various phosphoinositides. Kir 2.1 for example shows almost exclusive activation by PI(4,5)P$_2$ and no activation by PI(3,4)P$_2$ and PI(3,4,5)P$_3$[3]. All Kir channels show some level of selectivity for PI(4,5)P$_2$ over PI(3,4)P$_2$ and PI(3,4,5)P$_3$, with the exception of Kir 6.2 which is equally activated by those lipids[3]. This suggests that LC-CoA activation may proceed via the non-selective phosphoinositide binding site of Kir6.2 channels. Indeed mutations in Kir2.1 that reduced its selectivity for PI(4,5)P$_2$ converted this channel into a LC-CoA-activated channel[3]. Most members of the two pore potassium (K$_{2P}$) channel family are also modulated by both PI(4,5)P$_2$ and LC-CoA. Unlike Kir channels, PI(4,5)P$_2$ activates

some K$_{2P}$ channels and inhibits others, and LC-CoA shows a similar, yet not identical pattern[27].

TRPV5 and TRPV6 are Ca$^{2+}$ selective, inwardly rectifying, con-stitutively active ion channels. They share 75% sequence identity with each other, and less than 30% with other TRPV channels (TRPV1-4), which are outwardly rectifying, non-selective cation channels that are activated by high temperatures and various chemical ligands such as capsaicin for TRPV1. TRPV5 and TRPV6 are also very similar to each other functionally. Their constitutive activity in resting cells requires binding of PI(4,5)P$_2$ to these channels. PI(4,5)P$_2$ constitutes up the 1% the phospholipids in the plasma membrane, and it regulates many different ion channels, for most of which it serves as an obligate cofactor for activity[28,29]. Both TRPV5 and TRPV6 undergo Ca$^{2+}$-induced inactivation, which is mainly mediated by binding of Ca$^{2+}$-calmodulin (CaM) to the channel[30–33]. The structures of both TRPV5[18,34] and TRPV6[35] have been determined with CaM. These structures show that a single CaM binds to the channel and inhibits it by inserting K115 in CaM into the gating ring formed by W583 at the cytoplasmic end of S6. Depletion of PI(4,5)P$_2$ by Ca$^{2+}$-induced activation of phospholipase

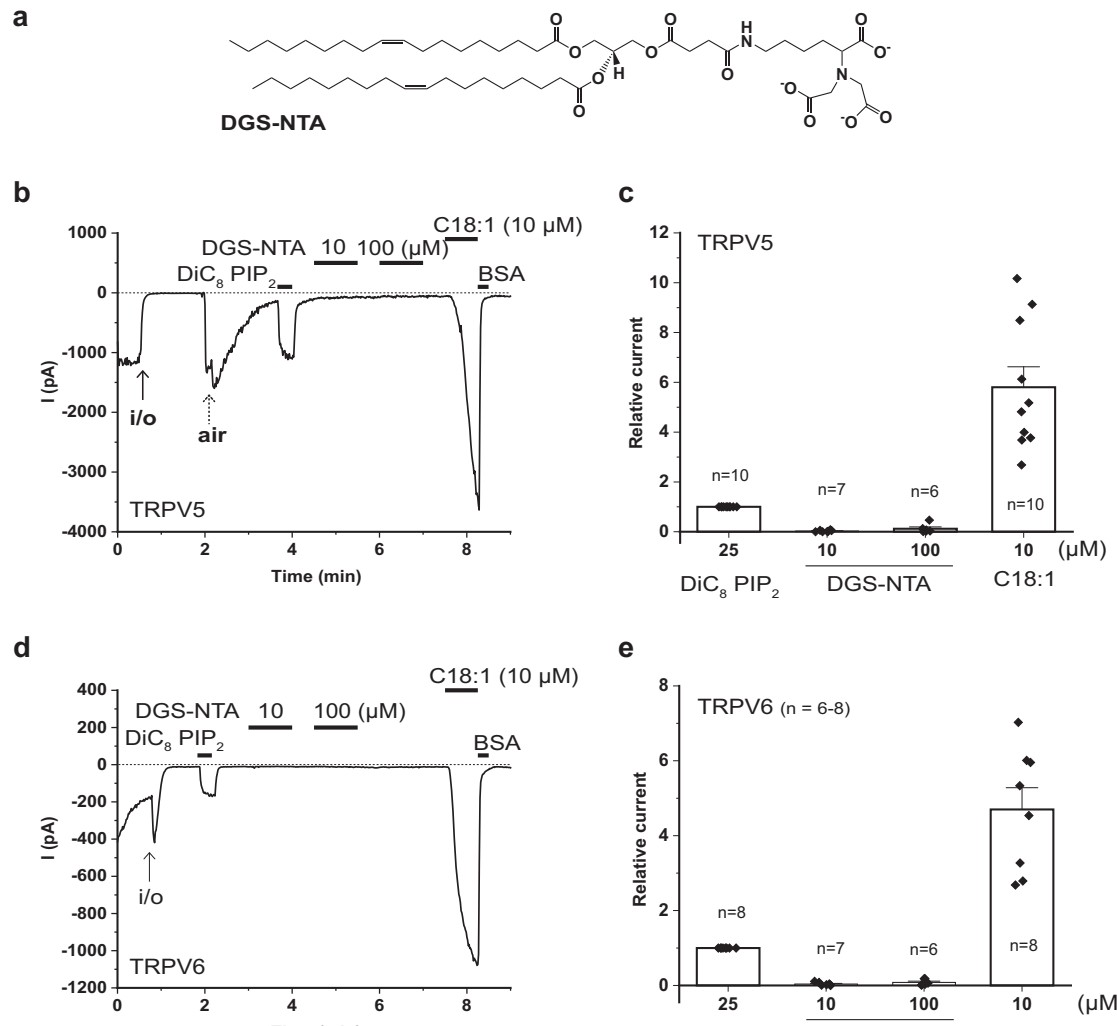

**Fig. 3 | DGS-NTA does not activate TRPV5 and TRPV6 channels. a** Chemical structure of DGS-NTA. **b-e** Excised inside-out patch clamp recordings on *Xenopus laevis* oocytes expressing TRPV5 and TRPV6 were performed as described in Methods. The establishment of the inside-out configuration is indicated by the arrows. In panel **b** an air bubble was applied to break the membrane vesicle. Traces show currents at −100 mV. Dashed lines show zero current. The applications of 25 µM diC$_8$ PI(4,5)P$_2$, and 10 and 100 µM DGS-NTA, and 10 µM oleoyl-CoA (C18:1) are shown by the horizontal lines. **b, d** Representative traces for the effect of diC$_8$ PI(4,5)P$_2$, DGS-NTA, and C18:1 CoA on TRPV5 (**b**) and TRPV6 (**d**). **c, e** Summary of the effects of diC$_8$ PI(4,5)P$_2$, DGS-NTA, and C18:1 CoA on TRPV5 (**c**) and TRPV6 (**e**). Bar graphs show mean ± SEM and scatter plots. The symbols represent experiments from individual oocytes.

C (PLC) enzymes has also been shown to contribute to Ca$^{2+}$-induced desensitization of TRPV6[17]. In a cellular environment, TRPV6 activity is likely determined by the balance between Ca$^{2+}$ influx and CaM binding to the channel keeping the channel partially closed to avoid toxic Ca$^{2+}$ overload. In native epithelial cells these channels are co-expressed with Ca$^{2+}$ binding proteins calbindins, that chelate Ca$^{2+}$ reducing inactivation and allowing more efficient transepithelial Ca$^{2+}$ transport[36].

How does LC-CoA come into this picture? We show here that LC-CoA can also maintain channel activity in a cellular environment when PI(4,5)P$_2$ is depleted. We showed this both by supplying LC-CoA through the whole-cell patch pipette (Fig. 6), and by increasing endogenous LC-CoA levels (Fig. 7). Modulation of K$_{ATP}$ by LC-CoA has been proposed to play roles in reduced insulin secretion in type 2 diabetes mellitus, and activation of cardiac K$_{ATP}$ during ischemia[37]. For TRPV6, one may speculate that the high glucose and high fatty acid concentrations we used to increase cytoplasmic LC-CoA levels, may occur in the duodenum, where TRPV6 is located, after consuming high fat and high carbohydrate diet. Since Ca$^{2+}$-induced activation of PLC and the ensuing depletion of PI(4,5)P$_2$ was shown to induce TRPV6 inactivation, and limit intestinal Ca$^{2+}$ transport[38], increased

intracellular LC-CoA levels, in principle, may modulate intestinal TRPV6 activity, by reducing reliance on PI(4,5)P$_2$ and thus increasing channel activity. High fat diet was indeed shown to increase intestinal Ca$^{2+}$ absorption, but whether this was due to increased TRPV6 activity was not examined[39,40]. It will require future studies to uncover how endogenous LC-CoA contributes to TRPV5 and TRPV6 channel activity in physiological or pathophysiological conditions. Such experiments may be made challenging by the difficulty of recording endogenous TRPV5 and TRPV6 activity from native tissues.

Our data provide structural insight into how LC-CoA opens TRPV5 and TRPV6 channels. Our cryoEM structures show that oleoyl-CoA binds to the same binding site as PI(4,5)P$_2$ in TRPV5, and it induces a conformational change very similar to that induced by PI(4,5)P$_2$. Our functional data show that diC$_8$ PI(4,5)P$_2$ did not further activate TRPV5 and TRPV6 after activation by LC-CoA, which is compatible with the two lipids acting at the same binding site. If the two lipids acted at different sites, one may have expected the two agents to have synergistic effects, such as menthol and PI(4,5)P$_2$ for TRPM8[16,41]. DiC$_8$ PI(4,5)P$_2$ not only failed to further activate TRPV5 and TRPV6, but induced a slight decrease in the currents evoked by LC-CoA, indicating

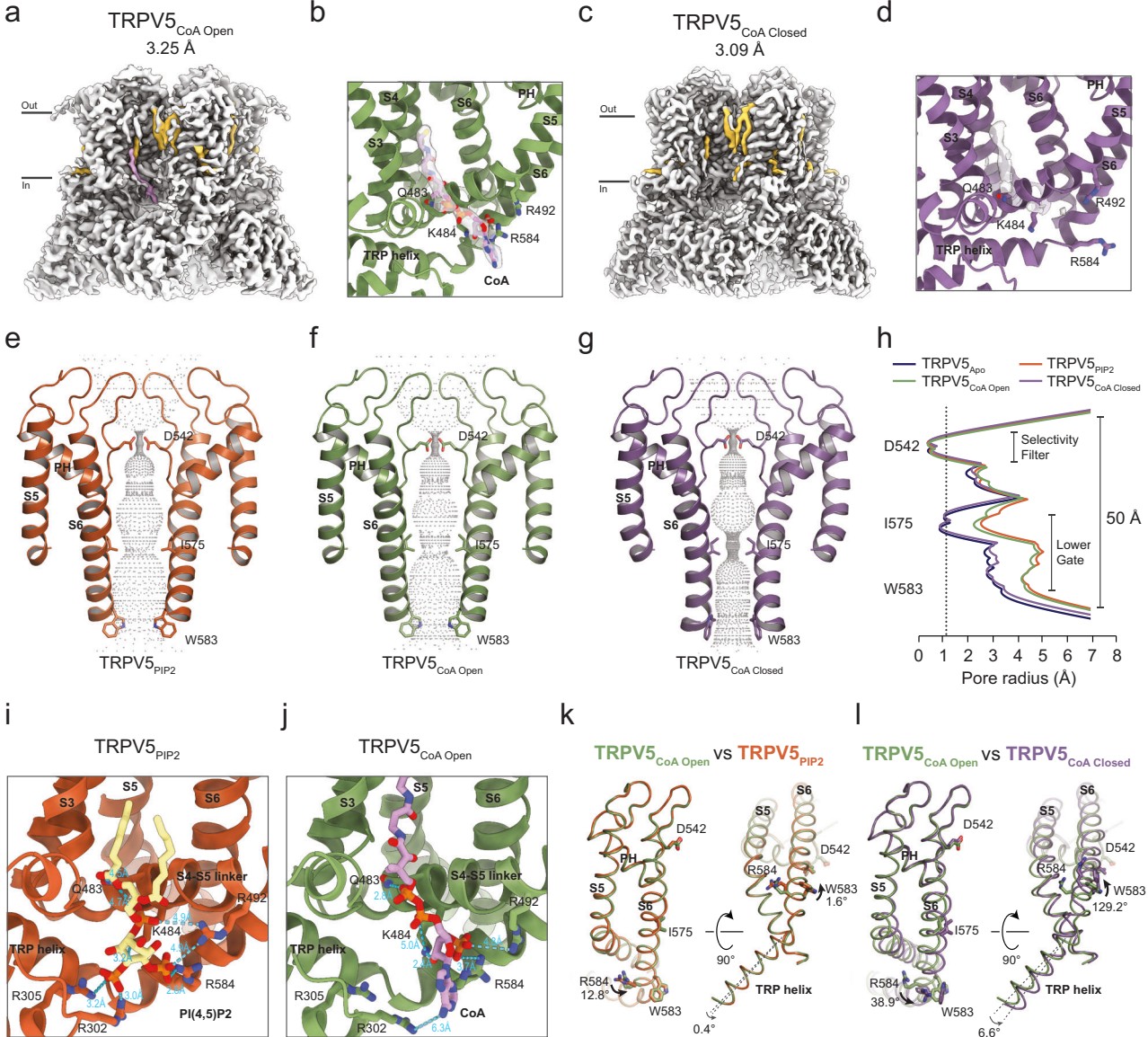

**Fig. 4 | Oleoyl-CoA binding site and conformational changes of TRPV5.** The TRPV5 CryoEM map in presence of oleoyl-CoA in open (**a**) and closed state (**c**) at a resolution of 3.25 Å and 3.09 Å, respectively. The CoA density is highlighted in pink and lipids in yellow. Coordinate model of CoA binding site in open (TRPV5$_{CoA Open}$) (**b**) and closed (TRPV5$_{CoA Closed}$) (**d**) state with overlaid density contoured at σ = 3.5. Pore profile of TRPV5 in presence of PI(4,5)P$_2$ (PDB 8FFO, TRPV5$_{PIP2}$) (**e**), TRPV5$_{CoA Open}$ (**f**), and TRPV5$_{CoA Closed}$ (**g**). **h** Pore radius along the conducting pathway of

TRPV5$_{Apo}$ (blue), TRPV5$_{PIP2}$ (orange), TRPV5$_{CoA Open}$ (green), and TRPV5$_{CoA Closed}$ (purple). Dotted gray line at 1.1 Å is the dehydrated calcium radius. Conservation of the PI(4,5)P$_2$ (**i**) and CoA (**j**) binding site. Distance between activator and the residues forming the binding site (R302, K484, R492, R584) are shown in blue. Comparison of pore domain movements (S5, S6 helixes) of TRPV5$_{CoA Open}$ *versus* TRPV5$_{PIP2}$ (**k**) and TRPV5$_{CoA Open}$ *versus* TRPV5$_{CoA Closed}$ (**l**) from the side (left) and the bottom (right) view.

that diC$_8$ PI(4,5)P$_2$ is a less efficacious agonist (partial agonist) compared to LC-CoA.

Our functional data show that opening of TRPV5 and TRPV6 requires the presence of a long acyl chain, and the 3'-phosphate. The dependence on the acyl chain length and phosphates for TRPV5 and TRPV6 activation by LC-CoA was very similar to that for K$_{ATP}$ activation and Kir2.1 inhibition[26]. The dependence on both the number of phosphates and the acyl chain length is also similar to the determinants of activation of most ion channels by phosphoinositides. Most PI(4,5)P$_2$-dependent ion channels are activated less by PI(4)P than by PI(4,5)P$_2$ and they require higher concentrations of shorter acyl chain phosphoinositides such as diC$_8$ PI(4,5)P$_2$ than long acyl chain natural PI(4,5)P$_2$. This dependence of the acyl chain length is likely due to the higher lipid / water partition coefficient of

longer acyl chain lipids, resulting in higher local concentrations in the membrane[21]. Interestingly, on K$_{2P}$ K$^+$ channels, LC-CoA species with double bonds had smaller effects[27], but we did not observe substantial difference between the effect by the presence of a double bond on TRPV5 and TRPV6.

Currently all open TRPV5 structures, without any activating mutations, have been determined in the presence of PI(4,5)P$_2$[18,19]. Our open TRPV5 structure with oleoyl-CoA shows this lipid in the same binding site as PI(4,5)P$_2$, which points to the central role of the phosphoinositide binding site in TRPV5 gating. Currently there is no TRPV6 structure with PI(4,5)P$_2$, even though computational and functional studies suggest a similar binding site[18]. Given the almost indistinguishable effect of LC-CoA on TRPV5 and TRPV6, it is likely that the two lipids act on the same binding site in TRPV6, but definitive

**Table 2 | Cryo-EM data collection, refinement and validation statistics**

| | TRPV5$_{CoA\ Closed}$ (EMD-29085, PDB 8FHH) | TRPV5$_{CoA\ Open}$ (EMD-29086, PDB 8FHI) | TRPV5$_{PIP2}$ (EMD-29049, PDB 8FFO) |
|---|---|---|---|
| **Data collection and processing** | | | |
| Magnification | ×105,000 | ×105,000 | ×105,000 |
| Voltage (kV) | 300 | 300 | 300 |
| Electron exposure (e–/Å$^2$) | 43 | 43 | 42 |
| Defocus range (µm) | −0.8 to −2.5 | −0.8 to −2.5 | −0.8 to −2.5 |
| Pixel size (Å) | 0.83 | 0.83 | 0.83 |
| Symmetry imposed | C4 | C4 | C4 |
| Initial particle images (no.) | 2,090,296 | 2,090,296 | 1,122,031 |
| Final particle images (no.) | 60,454 | 22,498 | 115,278 |
| Map resolution (Å) | 3.09 | 3.25 | 3.50 |
| FSC threshold | 0.143 | 0.143 | 0.143 |
| **Refinement** | | | |
| Initial model used (PDB code) | 7T6J | 7T6J | 7T6J |
| Model resolution (Å) | 3.09 | 3.25 | 3.50 |
| FSC threshold | 0.143 | 0.143 | 0.143 |
| Map sharpening $B$ factor (Å$^2$) | −112.6 | −103.7 | −154.2 |
| Model composition | | | |
| Non-hydrogen atoms | 19,780 | 20,116 | 20,100 |
| Protein residues | 2420 | 2440 | 2416 |
| Ligands | 12 | 16 | 28 |
| $B$ factors (Å$^2$) | | | |
| Protein | 58.66 | 81.44 | 56.52 |
| Ligand | 46.87 | 75.91 | 42.82 |
| R.m.s. deviations | | | |
| Bond lengths (Å) | 0.026 | 0.024 | 0.024 |
| Bond angles (°) | 1.843 | 1.713 | 1.711 |
| Validation | | | |
| MolProbity score | 1.22 | 1.38 | 1.22 |
| Clashscore | 4.41 | 5.36 | 4.45 |
| Poor rotamers (%) | 0.00 | 0.00 | 0.19 |
| Ramachandran plot | | | |
| Favored (%) | 98.00 | 97.53 | 99.00 |
| Allowed (%) | 2.00 | 2.47 | 1.00 |
| Disallowed (%) | 0.00 | 0.00 | 0.00 |

evidence for this requires determining co-structures of LC-CoA and PI(4,5)P$_2$ with TRPV6. In conclusion, our data provides structural insight into LC-CoA activation of an ion channel.

# Methods
## Lipids and chemicals
3'-Dephospho-Palmitoyl-CoA (cat. # NU-1180S) and 18:1 DGS-NTA (1,2-dioleoyl-sn-glycero-3-[(N-(5-amino-1-carboxypentyl)iminodiacetic acid)succinyl]) (cat. # 790528P) was purchased from Jena Bioscience and Avanti Polar Lipids, respectively. DiC$_8$ PI(4,5)P$_2$ (Ptdlns-(4,5)-P$_2$ (1,2-dioctanoyl)) (cat. # 64910) was obtained from the Cayman Chemical Company. Octanoyl CoA (cat. # O6877), Myristoyl CoA (cat. # M4414), Lauroyl CoA (cat. # L2659), Palmitoyl CoA (cat. # P9716), Stearoyl CoA (cat. # S0802), and Oleoyl-CoA (cat. # O1012) were

purchased from Sigma-Aldrich. Oleic acid-albumin from bovine serum (cat. # O3008), and the CPT1 inhibitor etomoxir (cat. # E1905) were purchased from Sigma. DiC$_8$ PI(4,5)P$_2$ was dissolved in bath solution at 2.5 mM, for excised patch measurements, and in pipette solution for whole cell patch clamp experiments. The various LC-CoA species were dissolved in bath solution at 5 or 10 mM, for the excised patch measurements, and in pipette solution for whole cell patch clamp experiments. Both the diC$_8$ PI(4,5)P$_2$ and LC-CoA stock solutions were aliquoted, and kept at −80 °C. On the day of the experiment aliquots of the stocks were thawed and diluted to working concentrations in the experimental solutions. One frozen aliquot was used only for one experiment, thus the stock solutions did not undergo repeated freezing and thawing.

## Mammalian cell culture and transfection
Human Embryonic Kidney 293 (HEK293) cells (CRL-1573) and HEK 293T cells (CRL-3216) were purchased from American Type Culture Collection (ATCC). HEK293 and HEK293T were maintained in Minimal Essential Medium (MEM, Gibco) and Dulbecco's Modified Eagle's medium (DMEM, Gibco), respectively, supplemented with 10% FBS and 100 U/mL penicillin/100 µg/mL streptomycin in an incubator at 37 °C in 5% CO$_2$. Cells were transiently co-transfected with cDNA encoding human myc-tagged TRPV6 in the pCMV-tag3A vector, and ci-VSP in an EGFP-IRES vector in a ratio of 1:1 using the Effectene transfection reagent (Qiagen) according to the manufacturer's protocol. After 24 h, transfected cells were plated on poly-L-lysine coated 12-mm round coverslips and used in experiments 48–72 h after transfection.

## *Xenopus laevis* oocyte preparation and injection
Animal procedures were performed by protocols approved by the Institutional Animal Care and Use Committee at Rutgers New Jersey Medical School. Oocytes were prepared as described earlier[42]. Briefly, frogs were anesthetized in 0.25% Tricane-S solution. Once the frog was unconscious, it was placed ventral side up on the ice. Following a small incision (~1 cm) on the lower abdomen, the skin and muscle were cut with sharp scissors. Sacs containing oocytes were collected by forceps and the frog was either sacrificed by removing the heart, or the incision was closed with a suture, and the frog was allowed to recover from surgery. The oocyte sacs were torn into smaller pieces using forceps, and digested in 0.1–0.2 mg/mL collagenase (Sigma) in OR2 solution (82.5 mM NaCl, 2 mM KCl, 1 mM MgCl$_2$, 5 mM HEPES, pH 7.4) rotating overnight in a 16 °C refrigerator. Defolliculated oocytes were washed multiple times in OR2 and kept at 16 °C in OR2 solution supplemented with 100 Units/mL penicillin/100 µg/mL streptomycin and 1.8 mM CaCl$_2$ (OR2+).

cDNA encoding the rabbit TRPV5 or human TRPV6 in pGEMSH vector was digested with NheI-HF at 37 °C. cRNA was transcribed from the linearized cDNA using mMesssage mMachine T7 Kit (Thermo Fisher). cRNA was microinjected into healthy oocytes using a nanoliter 2000 injector system (World Precision Instruments) and measurements were performed 24–48 h after injection.

## Electrophysiology
Excised inside-out patch clamp experiments were performed similar to that described earlier[42] using glass capillaries (4 in, OD 1.5 mm) (World Precision Instruments) of 0.7–1.1 MΩ resistance, filled with a solution containing 96 mM LiCl, 1 mM EGTA, 5 mM HEPES, pH 7.4 to allow monovalent inward currents. After formation of GΩ seals, the inside-out configuration was established, and currents were measured using an Axopatch 200B amplifier (Molecular Devices) using a ramp protocol from −100 mV to +100 mV applied once every second. The bath solution contained 96 mM KCl, 5 mM EGTA, 10 mM HEPES, pH 7.4. Recording was performed at 18–20 °C. diC$_8$ PI(4,5)P$_2$ and various fatty acyl-CoA solutions were dissolved in bath solution and applied to the intracellular side of the patch membrane using a gravity-driven

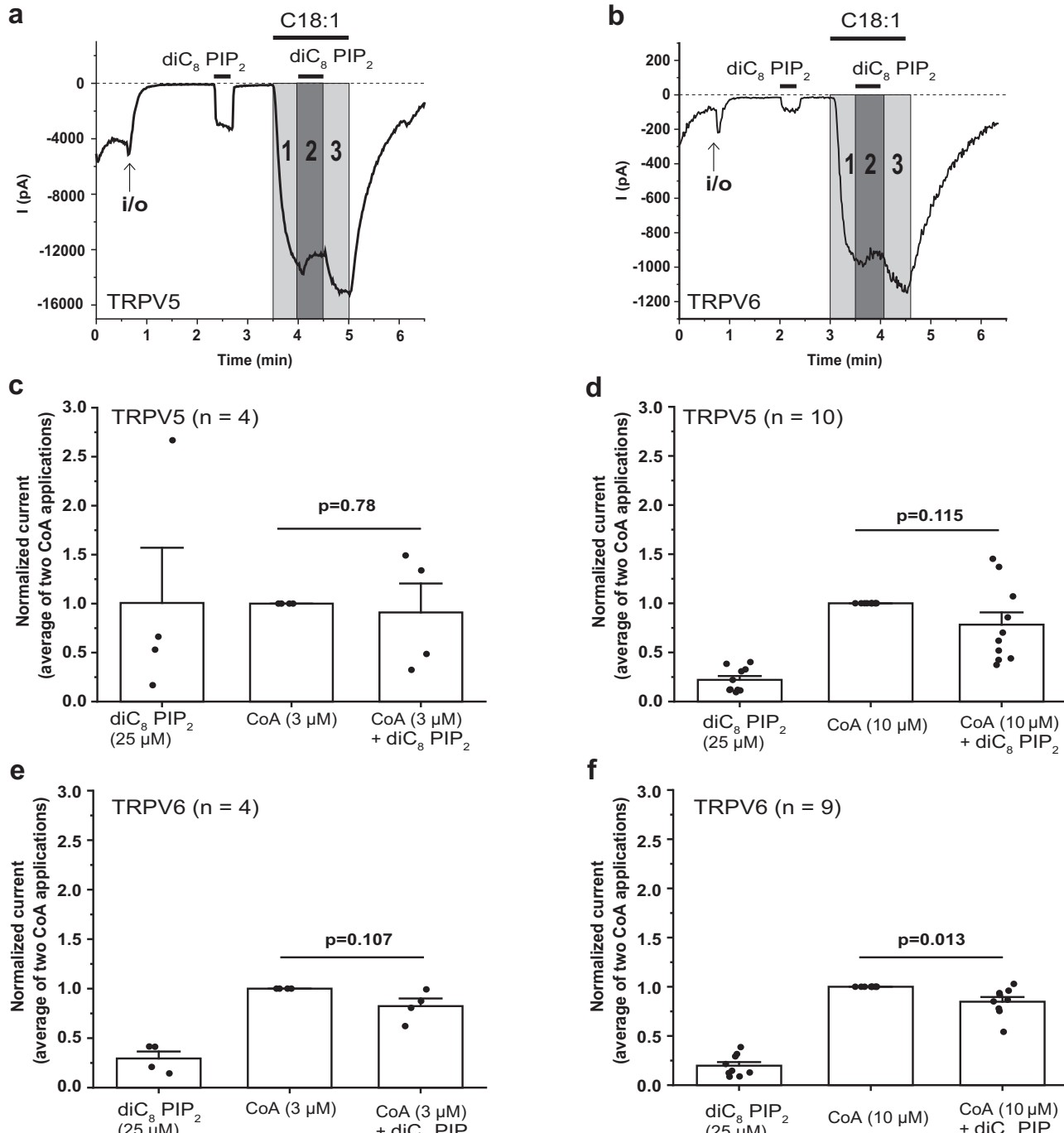

**Fig. 5 | The effect of co-application of LC-CoA and PI(4,5)P$_2$ on TRPV5 and TRPV6 channels.** Excised inside-out patch clamp recordings on *Xenopus* oocytes expressing TRPV5 (**a**, **c**, **d**) and TRPV6 (**b**, **e**, **f**) were performed as described in Methods. Traces show currents at −100 mV. Dashed lines show zero current. The establishment of the inside-out configuration is indicated by the arrows. The application of 25 μM diC$_8$ PI(4,5)P$_2$ and C18:1 CoA are shown by the horizontal lines. Representative trace shows the effect of co-application of 10 μM C18:1 CoA and 25 μM diC$_8$ PI(4,5)P$_2$ (PIP$_2$) on TRPV5 (**a**) and TRPV6 (**b**). Each number in the gray area indicates a different application solution; 1 and 3 are applications of C18:1 CoA and 2 is a co-application of C18:1 CoA and diC$_8$ PI(4,5)P$_2$. Summary of the data from 3 μM C18:1 CoA (**c**, **e**) and 10 μM C18:1 CoA (**d**, **f**) in the presence or in the absence of 25 μM diC$_8$ PI(4,5)P$_2$. In each panel, the middle bar indicates the average current at gray areas 1 and 3 (current from the application of C18:1 CoA). *P* < 0.05 (Paired Sample *t*-test, two-tailed). Bar graphs show mean ± SEM and scatter plots. The symbols represent experiments from individual oocytes.

perfusion system (ALA Scientific). Data were collected and analyzed with pCLAMP (Molecular Devices), and further analyzed and plotted with Origin 8.0 or Origin 2023b.

Whole-cell patch clamp recordings were performed on TRPV6 and ci-VSP-expressing HEK293 cells. The standard extracellular solution contained 142 mM LiCl, 1 mM MgCl$_2$, 10 mM HEPES, 10 mM Glucose,

pH 7.4. In this solution TRPV6 currents are blocked by Mg$^{2+}$ and trace amounts of Ca$^{2+}$. Monovalent currents were initiated by removing Mg$^{2+}$, and supplementing the solution with 1 mM EGTA, as described earlier[43]. Using LiCl as the charge carrier eliminates contribution of endogenous currents in Ca$^{2+}$ and Mg$^{2+}$ free conditions. For the low glucose conditions in Fig. 7c, d, glucose was replaced by 10 mM

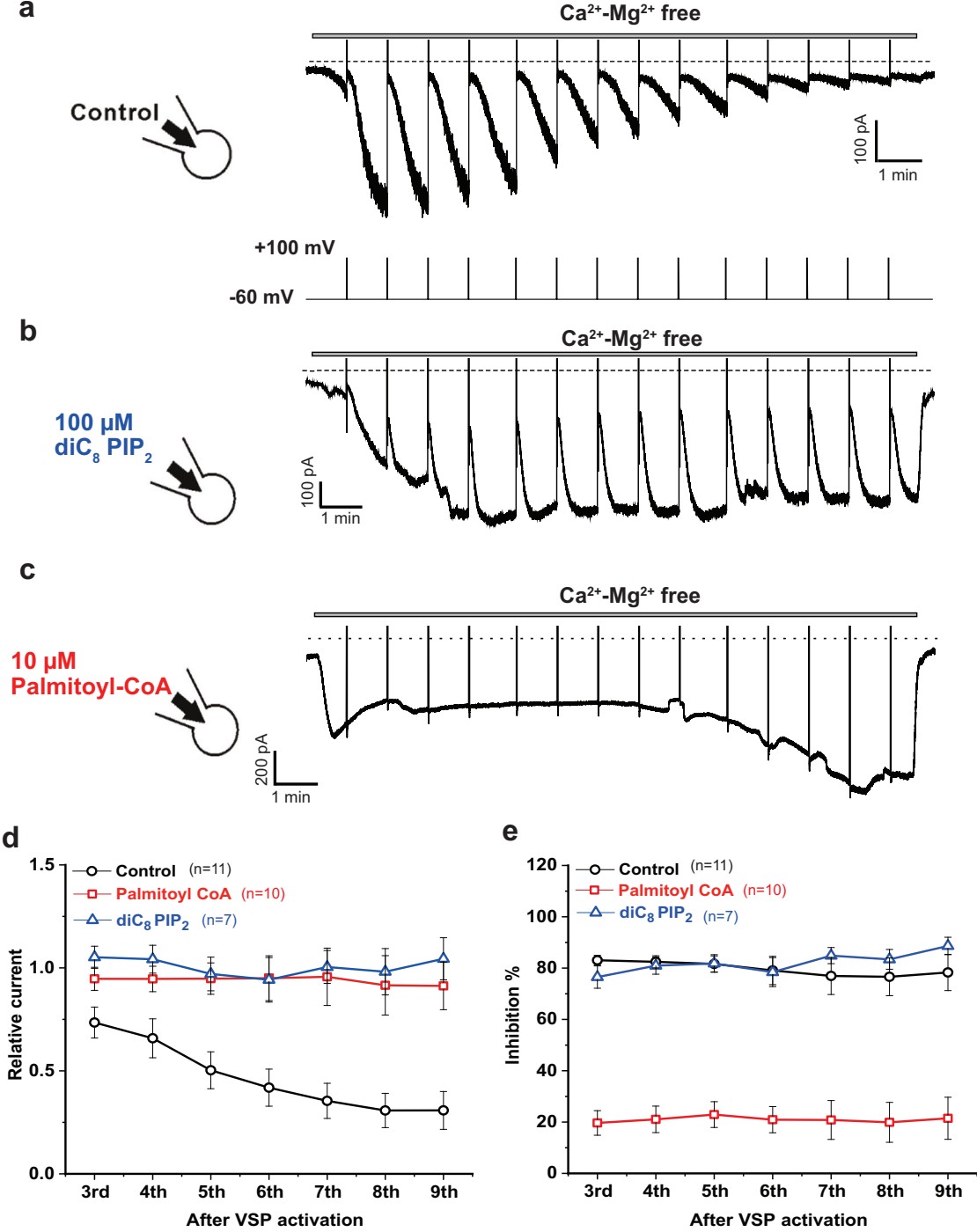

**Fig. 6 | Palmitoyl-CoA activates TRPV6 channels after PI(4,5)P$_2$ depletion in intact cells.** Whole-cell voltage-clamp recordings in HEK293 cells transiently cotransfected with TRPV6 and ci-VSP were performed as described in Methods. In these measurements, constant holding at −60 mV for 59 s, then a depolarizing step to +100 mV for 1 s was applied to activate ci-VSP and induce PI(4,5)P$_2$ hydrolysis. This was repeated up to 14 times continuously (**a**, bottom). The intracellular pipette solution did not contain ATP. **a** monovalent TRPV6 currents were measured by application of Mg$^{2+}$- and Ca$^{2+}$-free solution, as described in the methods section. Standard intracellular solution was supplemented with 100 μM diC$_8$ PI(4,5)P$_2$ (**b**) or 10 μM palmitoyl-CoA (**c**). **d** Summary of relative recovered current after +100 mV pulse compared to peak current. **e** Summary of the ratio between the steady-state current before the +100 mV pulse and inhibited current right after the +100 mV pulse. Line graphs in d and e show mean ± SEM from individual cells.

mannitol in the extracellular solution. The pipette (intracellular) solution used in recordings contained 140 mM potassium gluconate, 10 mM HEPES, 5 mM EGTA, 2 mM MgCl$_2$, pH 7.3. The pipette solutions were supplemented with 100 μM diC$_8$ PI(4,5)P$_2$ or 10 μM palmitoyl-CoA for Fig. 6b, c. Patch pipettes were pulled from borosilicate glass with filament (O.D. 1.5 mm, I.D. 0.75 mm, Sutter Instrument) and pipette resistance was 2−4 MΩ. Currents were measured using an Axopatch

200B amplifier (Molecular Devices). After formation of GΩ seals, the whole-cell configuration was established. The currents were measured at holding potential of −60 mV for 59 s then a depolarizing step to +100 mV for 1 sec was applied to activate ci-VSP to dephosphorylate PI(4,5)P$_2$. Currents were low-pass filtered at 2 kHz, and digitized using a Digidata 1322 A unit (Axon Instruments). No series resistance compensation was performed. Solutions were exchanged by gravity-driven

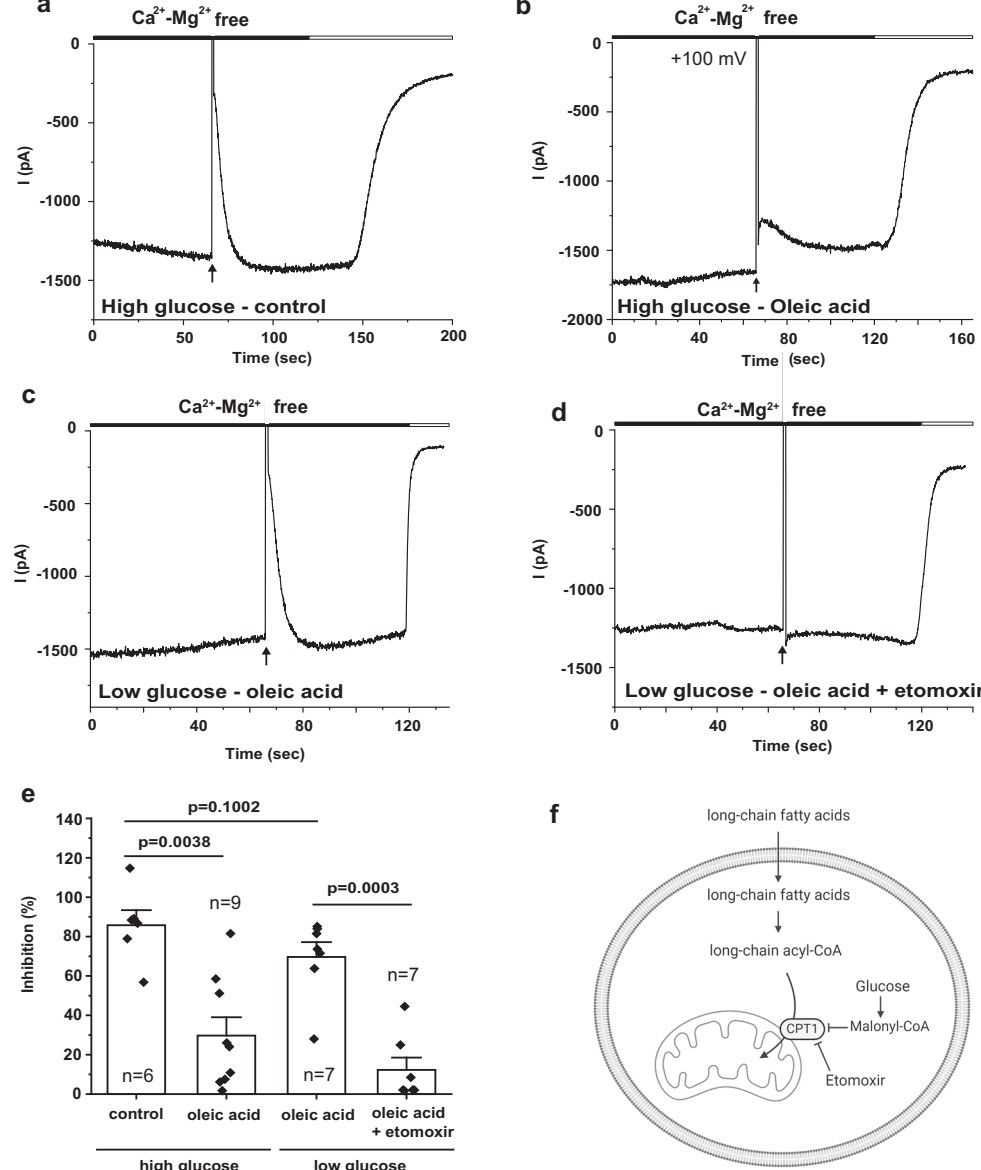

**Fig. 7 | Endogenous LC-CoA maintains TRPV6 activity. a-d** Representative traces for whole-cell patch-clamp recordings in HEK293T cells transiently cotransfected with TRPV6 and ci-VSP, performed as described in the Methods section. The membrane potential was clamped at −60 mV for 59 s, then a depolarizing step to +100 mV for 1 s was applied (arrows) to activate ci-VSP, after which the holding potential returned to −160 mV. Monovalent currents were measured in a Ca²⁺-Mg²⁺-free solution, and at the end of the experiment a solution containing 1 mM Mg²⁺ was applied to inhibit monovalent currents, see methods for details. The transfected cells were pretreated with (**b**) or without (**a**) 164 μM oleic acid overnight in DMEM containing high glucose (25 mM). For the low glucose condition, transfected cells were pretreated with 164 μM oleic acid (**c**) or 164 μM oleic acid with 100 μM etomoxir (**d**) overnight in the DMEM containing normal glucose (5.5 mM). **e** Summary of the inhibited currents right after the +100 mV pulse. Statistical significance was calculated with one way analysis of variance, overall ANOVA $F = 16.53$, $p = 3.96 \times 10^{-6}$. **f** Cartoon explaining the various treatments in these experiments. Long chain fatty acids (oleic acid) are taken up by the cell and converted to LC-CoA, which is transported from the cytoplasm to the mitochondria by CPT-1. High glucose, and etomoxir inhibits CPT-1, leading to increased cytoplasmic LC-CoA levels (created with Biorender). Bar graph shows mean ± SEM and scatter plots. The symbols represent experiments from individual cells.

---

perfusion system (ALA Scientific). Data were collected and analyzed with pCLAMP, and further analyzed and plotted with Origin 8.0, or Origin Pro 2023b.

## Statistics

Data are shown as the mean ± SEM and scatter plots and the number of measurements are shown in the figure or figure legend. A measurement in each oocyte, or cell is represented as one data point. Most experiments were performed on oocytes prepared from at least two independent isolations, or different cell transfections, see source data

for details. Statistical significance was calculated using paired $t$-test (two-tailed), or one way analysis of variance, as indicated in the figure legends in Origin. Normal distribution of the data was tested using the Kolmogorov Smirnov test in Origin.

## Protein expression and purification

Full length rabbit TRPV5 was prepared as previously described[44]. In brief, rabbit TRPV5 with a C-terminal 1D4 epitope tag in a YepM vector[45] was transfected into BJ5457 *Saccharomyces cerevisiae* (ATCC) using an alkali-cation yeast transformation kit (MP Biomedicals)

according to the manufacturer's instruction. Yeast expressing TRPV5 were resuspended in homogenization buffer (25 mM Tris-HCl, pH 8.0, 300 mM Sucrose, 5 mM EDTA, and protease inhibitor cocktail) and lysed using a M110Y microfluidizer (Microfluidics) at 100 psi. The membranes were isolated from cellular debris using three rounds of centrifugation (3000 × $g$ for 10 min, 14,000 × $g$ for 35 min, and 100,000 × $g$ for 1 h).

TRPV5 membranes were solubilized in 20 mM HEPES, pH 8.0, 150 mM NaCl, 2 mM TCEP, 10% glycerol, 1 mM PMSF, and 0.087% mM LMNG. The insoluble fraction was separated by centrifugation at 100,000 × $g$ for 1 h. The supernatant was incubated with 1D4-antibody coupled to CnBr-activated Sepharose beads for 3 h. The beads were collected on a gravity flow column and washed with wash buffer containing 20 mM HEPES, pH 8.0, 150 mM NaCl, 2 mM TCEP, and 0.006% mM DMNG. The beads were incubated with elution buffer (20 mM HEPES, pH 8.0, 150 mM NaCl, 2 mM TCEP, and 0.006% mM DMNG, 3 mg/mL 1D4 peptide) overnight. TRPV5 was eluted once every 5 min in ten 1 mL fractions. Peak fractions were pooled and concentrated at 2.88 mg/ml (0.75 ml) using a 100-kDa concentrator (Millipore).

TRPV5 was reconstituted into nanodiscs in a ratio of TRPV5:MSP2N2:Lipids:DMNG of 1:1:200:500. Lipids (soy polar lipids, Avanti) were dried under a nitrogen flow and stored under vacuum until they were resuspended in the detergent free wash buffer with the DMNG in a 1:2.5 ratio (Lipids:DMNG). The assembled nanodisc reconstitution mixture was incubated at 4° for 30 min, then about 30 µL of Bio-Beads were added before rotating at 4 °C for 1 h. The mixture was transferred to a new tube with 30 µL of fresh Bio-Beads and incubated in rotation overnight at 4 °C. Finally, the reconstituted TRPV5 into nanodiscs was run on a Superose 6 increase 10/300 GL column (GE Healthcare) equilibrated with 20 mM HEPES, pH 8.0, 150 mM NaCl, 2 mM TCEP. Fractions containing nanodisc-reconstituted TRPV5 were pooled, concentrated at 2.7 mg/ml, and used for grid preparation.

### Cryo-EM sample preparation and data collection, data processing and model building

The TRPV5 nanodisc sample was incubated with 400 µM Oleoyl-coenzyme A lithium salt (Sigma Aldrich) (TRPV5$_{CoA}$) or 400 µM diC$_8$-PI(4,5)P$_2$ (Echelon Biosciences) for 45 min. Using a Vitrobot Mark IV (Thermo Scientific), 3 µL of the samples was applied onto glow-discharged 1.2/1.3 Quantifoil Holey Carbon Grids (Quantifoil Micro Tools) for TRPV5$_{CoA}$ and ANTCryo (Single Particle) for TRPV5$_{PIP2}$ preparation. The sample was blotted in a chamber at 4 °C and 100% humidity for 6–9 s with 0 blot force before being frozen in liquid ethane.

Samples were imaged on a Titan Krios G3i 300 kV electron microscope with a Gatan K3 direct electron detector. 42 frame movies were collected with a nominal dose of 42 e⁻/Å². Images datasets were collected at a magnification of 105,000 and a pixel size of 0.43 Å/pixel.

For the TRPV5$_{PIP2}$ dataset, 5,216 movies were collected and processed using cryoSPARC v4.0.1[46–48]. Movies were patch motion corrected with an alignment resolution of 2 Å and with a Fourier-crop factor of 0.5, then run through patch CTF estimation. Using manual curate exposure with an estimated maximum resolution below 8 Å, 409 micrographs were rejected. 97,366 particles were blob picked from the subsets of 300 micrographs and used to create a template for autopicking from the whole dataset. 1.2 million particles were extracted and binned by a factor of 4 with a box size of 72 pixels. These particles were 2D-classified with 50 classes, resulting in 600,674 good particles that were re-extracted and binned by a factor of 2 with a box size of 144 pixels. Particles were oriented with the TRPV5 apo map (EMD-25716) and refined with a non-uniform refinement in C1 symmetry to generate the initial model. A three-class heterogenous refinement was run to separate good particles, resulting in a class of 503,915 particles. Particles were unbinned and re-extracted with a box

size of 288 pixels, then were subjected to 3D classification with 12 classes focusing on the pore region. Four classes with well-defined pore region were obtained; three of them with a strong density at the putative PI(4,5)P$_2$ site. These classes with 115,728 particles were combined and refined with a non-uniform refinement to build the final models in C1 and C4 symmetry at a resolution of 3.7 Å and 3.5 Å, respectively.

TRPV5$_{CoA}$ datasets were processed using cryoSPARC v3.3.2 and v4.0.2;[46–48] 11,853 movies were collected. Movies were patch motion corrected with an alignment resolution of 5 Å and with a Fourier-crop factor of 0.5, then run through patch CTF estimation. 1582 movies were rejected using manual curate exposure with an estimated max resolution below 8 Å. 26,460 particles were autopicked from a subset of 200 micrographs and used as template to process the full accepted micrographs. 2.3 millions of particles were extracted and binned by a factor of 4 with a box size of 72 pixels, then sorted with three rounds of 2D classification on 50 classes: 374,895 particles were reextracted and binned by a factor of 2 with a box size of 144 pixels. Particles were oriented with TRPV5 apo map (25716) and refined with Non-uniform (NU) refinement in C4 symmetry (3.44 Å) to generate the first model, then 3D classification was performed on 10 classes. 4 classed at the highest resolution, three of them in the same closed conformational stated and one in the open state, were reextracted in an unbinned box size of 288 pixels, and used to build the final models in C4 symmetry. The TRPV5$_{CoA\ Open}$ map was built with 22,498 particles at a resolution of 3.25 Å and with 60,454 at 3.09 Å for the TRPV5$_{CoA\ Closed}$.

The TRPV5 apo state model (7T6J)[19] was used as an initial model, and adjusted manually in COOT[49], then lipids, Oleoyl-coenzyme A (3VV), obtained from ePDB database, were docked and adjusted manually. Real Space_Refinement, and eLBOW from the PHENIX software were used to refinement and to generate ligand restraints[50,51], respectively. Chimera, ChimeraX, and PyMol were used to analyze, visualize, and to generate figures[52–54]. Pore profiles were determined using HOLE[55].

### Reporting summary

Further information on research design is available in the Nature Portfolio Reporting Summary linked to this article.

## Data availability

The data that support this study are available from the corresponding authors upon request. The atomic coordinates and cryo-EM density maps of the structures presented in this paper are deposited in the Protein Data Bank and Electron Microscopy Data Bank under the accession codes 8FFO and EMD-29049 (TRPV5$_{PIP2}$), 8FHI and EMD-29086 (TRPV5$_{CoA\ Open}$), and 8FHH and EMD-29085 (TRPV5$_{CoA\ Closed}$), respectively. Source data are provided with this paper.

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

## Acknowledgements

This work was supported by National Institute of Health grants (R01GM093290 to T.R and R35GM144120 to V.Y.M.-B.) and a National Research Foundation of Korea (NRF) grant funded by the Korea government (MSIT) (Nos. RS-2023-00212567 and RS-2023-00219399) to B.H.L. We also acknowledge the use of instruments at the Electron Microscopy Resource Lab and at the Beckman Center for Cryo-Electron Microscopy at the University of Pennsylvania and thank Stefan Steimle for assistance with the Krios microscope.

## Author contributions

B.H.L. and T.R. conceived and designed the electrophysiology experiments, B.H.L. performed the electrophysiology experiments. J.J.d.J.-P., and V.Y.M.-B. conceived and designed the CryoEM experiments. J.J.d.J.-P purified and prepared the cryo-EM samples of TRPV5, and processed the cryo-EM data and built the structural models for TRPV5. T.R and V.Y.M.-B. supervised the electrophysiology and CryoEM experiments, respectively. B.H.L., and J.J.d.J-P visualized the data. B.H.L., J.J.d.J-P, T.R. and V.Y.M.-B drafted and edited the manuscript, and B.H.L and T.R. finalized the manuscript.

## Competing interests

The authors declare no competing interests
