## [Peer Review File · Nature Communications]

Structural basis of the activation of TRPV5 channels by long-chain acyl-Coenzyme-AReviewers' Comments:

Reviewer #1:

Remarks to the Author:

In this study Lee et al focus on the role of acyl-2 coenzyme-A in the function of the constitutively active ion channels Transient Receptor Potential Vanilloid 5 and 6 (TRPV5 and TRPV6). Their findings show that long chain acyl-2 coenzyme-A (LC-CoA) induce channel activation, in a manner similar to that observed for PIP2. However, while TRPV5 and TRPV6 activation by PIP2 can be achieved even by short-chained, water-soluble analogs (implying a strong dependency on the headgroup and less dependency on the acyl chain length), activation of TRPV5 and TRPV6 channels by CoA required both a long acyl chain and the 3' phosphate in the LC CoA head group.

The authors also present cryo-EM data which shows that LC CoA binds in the PIP2 binding site in TRPV5. Furthermore, the open state of the channel correlates with the presence of the presumed LC CoA density. The mechanism of channel opening by LC CoA is also analogous to the one previously identified for PIP2.

Overall, this is a nice study, and the results will likely be of interest to both the TRP channel and the wider ion channel community.

I have these minor comments and questions:

In the traces shown in Fig1 none of the currents elicited by CoAs reach steady state. This may end up misrepresenting the effects of these molecules. In fact, most of the traces of CoA currents shown in the paper do not reach steady state. The authors should comment on this.

In Fig 1e (TRPV6 response to C18:1), the currents are not allowed to reach baseline before addition of the next, higher concentration of the compound. It is difficult to discern if the effect is additive (5 μ M + 10 μ M) or purely due to the application of 10 μ M. This also complicates the interpretation of data shown in Fig 1f.

Here the authors aim to show that PIP2 and LC CoA use the same binding site and mechanism to activate the channel. I think this is a fair hypothesis based on their data. However, I do not fully understand the approach.

The authors apply LC CoA to get a large TRPV5/TRPV6 current, then apply the soluble PIP2 analog at a concentration that produces a submaximal response to detect if the current increases.

My comments to this are:

1) diC8 PIP2 at the 25 μ M concentration on its own produces a small current compared to the currents induced by LC CoAs. Could you expect to detect a potential current increase following PIP2 application? Especially given that we again do not reach steady state?

2) Why do the authors choose a submaximal concentration of PIP2 for this experiment?

Given how similarly the two molecules appear to act on TRPV5 and TRPV6, can the authors offer any insights into why the length of the acyl chain does not seem to impact activation by PIP2 but is critical for acyl-CoA?

Standardize y axis in Fig 5 panels b-e.

In the methods, can the authors describe how the lipid stocks were prepared and handled?

I think the manuscript could do with a few supplementary items, like a figure describing the cryo-EM data collection and processing, local resolution, density coverage in critical parts of the channel and figures showing additional traces (e.g., application of different acyl chain length CoAs in Fig1, TRPV6 traces in Fig5....)

I also found a few typos:

163 the TRPV5CoA Closed structure do not have such additional density / does

165 (S6 helix) (Fig. 4b), the same residues that was reported for the PI(4,5)P2-binding site / were

408 1:1:200:500. Lipids (soy polar lipids, Avanti) were dried under a nitrogen flow and store / stored

409 under vacuum until were resuspended in the detergent free wash buffer with the DMNG / they were

417 used to grid preparation. / used for

421 TRPV5 into nanodiscs sample was incubated / TRPV5 nanodisc sample

423 Scientific), 3 μ L of the samples were applied onto glow-discharged 1.2/1.3 Quantifoil / 3 μ L of the samples was

435 estimated max resolution below to 8 Å. / below 8 Å

436 of 200 micrographs and used as templated / template

440 Particles were oriented with TRPV5 apo map (25716) and refine with Non-uniform (NU) / refined

441 refinement in C4 symmetry (3.44 Å) to generate the first model, then was performed 3D / then 3D was performed

Reviewer #2:

Remarks to the Author:

LC-CoA esters are metabolic cofactors produced in certain cell and tissue types that have been shown to regulate a host of cellular processes, including modulation of ion channel activity. Here, the authors report TRPV5 and TRPV6 are activated by CoA species exhibiting long acyl chains and a 3' phosphate. The authors also solved the Cryo-EM structure of TRPV5 bound to an LC-CoA, which reveals that LC-CoA binds the same site as PI(4,5)P2 identified in their prior structures. When LC-CoA and the soluble PI(4,5)P2 analog DiC8 PIP2 are sequentially co-applied in excised patches, the authors do not observe a change in activation, which they interpret as LC-CoA occluding or preventing binding of DiC8 PIP2. Whole-cell patch experiments show that exogenous addition of LC-CoA is also able to maintain TRPV6 activity when PI(4,5)P2 is depleted from the membrane with VSP. The manuscript and its findings will be greatly elevated by addressing the following points:

Address physiological relevance: What is the evidence that the LC-CoAs used in this study are produced in the same cell types that endogenously express TRPV5/6 (currently there is a loose mention that TRPV5/6 may be in some pancreatic cells)? Are the LC-CoAs produced at a sufficient concentration in the plasma membrane (e.g., at levels comparable to those used in this manuscript) for this interaction to be physiologically relevant? If not, then the authors should shift their focus and make a compelling case for LC-CoA as a research tool and demonstrate its usefulness over PI(4,5)P2. Either way, a cell line that is known to endogenously produce LC-CoAs would be ideal for the authors to use for a similar experiment to Figure 6. This would speak to whether endogenous levels of LC-CoAs are capable of gating TRPV5/6 since the channel would presumably retain activity if the LC-CoA is not affected by VSP. If this experiment is not feasible and/or it does not work, the authors should still address it.

Contextualize and clarify existing results: The authors should better contextualize their structural data with their functional data. Currently, discussion of the lipid binding site is limited to residues conserved in PIP2 binding, which predominantly engage the lipid headgroup. However, structural revelations about the importance of the acyl chain length or fatty acid saturation are not discussed. The authors should also discuss how their structure contributes to understanding the reversibility of LC-CoA binding as opposed to the nearly irreversible binding of PIP2. Additionally, the results in Figure 5 would benefit from a more critical analysis. At a saturating (10 μ M) concentration of LC-CoA, the authors see a dip in TRPV5/6 activity when diC8 PIP2 is applied, which is significant for TRPV6. This could be explained by exchange of the better activating LC-CoA for the weaker activating DiC8 PIP2 and the authors should better interpret this dip. Given this, it is unclear why the authors observe no

change in TRPV5/6 activity when a sub-saturating (3 μ M) concentration of LC-CoA is co-applied with DiC8 PIP2. The authors should also address whether PIP2 instead of DiC8 PIP2 is a better reagent for this experiment - since PIP2 may have increased binding affinity and may be more comparable to LC-CoA in its activity. Finally, the authors should better explain how their data supports their conclusion that LC-CoA "occludes" PIP2. The authors propose both LC-CoA and PIP2 activate TRPV5/6 by binding the same site, thus they have the same effect on channel function, so perhaps "substitutes for" or "replaces" would be more accurate than "occludes".

Additional data/experiments that are needed to strengthen their findings:

1. The authors use DiC8 PIP2, a soluble PIP2 analogue, throughout their manuscript to make direct comparisons between PIP2 and LC-CoAs. The authors should directly compare TRPV5/6 activation by PIP2 to activation by diC8 PIP2 (and distinguish the diC8 PIP2 from endogenous PIP2 in their figures - current labeling is misleading). This will allow them to scale or calibrate their comparative results from LC-CoAs and diC8 PIP2. This scaling control is important because the authors appear to assume that the soluble short chain PIP2 analogue (diC8 PIP2) and PIP2 are equally capable of activating TRPV5/6, however, these lipids cannot be equal based on the authors' findings with LC-CoAs. LC-CoAs and PIP2 occupy the same site in TRPV5/6 and the authors show there is a strong effect of alkyl chain length on LC-CoA activation. DiC8 PIP2 has an alkyl chain length (C8) that was nonfunctional in an LC-CoA, which may explain why they needed a higher concentration of diC8 PIP2 than LC-CoA to activate TRPV5/6.
2. This study would benefit from validation of the TRPV5 structures with mutagenesis to disrupt LC-CoA binding residues, followed by functional analysis. Can they make mutations that preclude LC-CoA binding, but not PIP2? While the authors were unable to solve the structure of TRPV6 bound to LC-CoA, the authors may be able to apply the same mutagenesis strategy to TRPV6 to strengthen their claim that TRPV6 uses a similar binding mechanism for LC-CoA as TRPV5.
3. To test the relative ability of LC-CoA and PIP2 each to displace the other, the authors should collect data where they activate with Lipid 1, coapply Lipids 1 and 2, then remove Lipid 1. The slope of the change in current from Lipid 1+2 to Lipid 2 would give the authors the displacement rate. These experiments should be performed with both LC-CoA and diC8 PIP2 as Lipid 1, allowing a more quantitative comparison of displacement rates to determine whether LC-CoA indeed has a higher affinity for the binding site and "occludes" PIP2, as the authors claim.

Response to the reviewers' comments:

REVIEWER COMMENTS

Reviewer #1 (Remarks to the Author):

In this study Lee et al focus on the role of acyl-2 coenzyme-A in the function of the constitutively active ion channels Transient Receptor Potential Vanilloid 5 and 6 (TRPV5 and TRPV6). Their findings show that long chain acyl-2 coenzyme-A (LC-CoA) induce channel activation, in a manner similar to that observed for PIP2. However, while TRPV5 and TRPV6 activation by PIP2 can be achieved even by short-chained, water-soluble analogs (implying a strong dependency on the headgroup and less dependency on the acyl chain length), activation of TRPV5 and TRPV6 channels by CoA required both a long acyl chain and the 3' phosphate in the LC CoA head group.

The authors also present cryo-EM data which shows that LC CoA binds in the PIP2 binding site in TRPV5. Furthermore, the open state of the channel correlates with the presence of the presumed LC CoA density. The mechanism of channel opening by LC CoA is also analogous to the one previously identified for PIP2.

Overall, this is a nice study, and the results will likely be of interest to both the TRP channel and the wider ion channel community.

Thank you for the positive evaluation

I have these minor comments and questions:

In the traces shown in Fig1 none of the currents elicited by CoAs reach steady state. This may end up misrepresenting the effects of these molecules. In fact, most of the traces of CoA currents shown in the paper do not reach steady state. The authors should comment on this.

Thank you for the comment. Applying LC-CoA to excised patches often makes the patches unstable. In most cases we applied short pulses to avoid patch rupture, and demonstrate reversibility. Full steady state is difficult to achieve, as the initial fast increase is usually followed by a slower long-lasting increase, as shown in Figure 5A. This is similar to the application of long acyl chain PI(4,5)P₂, where TRPV6 currents keep increasing for several minutes, see our earlier publications Zakharian et al FASEB J 2011 (Fig. 6A there) PMID 21810903 and Velisetty et al 2016 (Fig. 2, Fig. 5. there), PMID: 27291418. We commented on this in the revised manuscript on page 3.

In Fig 1e (TRPV6 response to C18:1), the currents are not allowed to reach baseline before addition of the next, higher concentration of the compound. It is difficult to discern if the effect is additive (5 uM + 10 uM) or purely due to the application of 10 uM. This also complicates the interpretation of data shown in Fig 1f.

Performing, and quantifying dose-response measurements with slowly acting lipophilic

compounds that accumulate in the membrane is inherently difficult, and there is no perfect, yet practical way to do it. In the heart of the problem is that we only control the concentration of the lipid in the aqueous solution, but we do not know the concentration of the lipid in the membrane. It is not even clear how to define concentration of an amphipathic lipid in the context of the plasma membrane, where it essentially moves in two dimensions, in a relatively fixed orientation, presumably the negatively charged head group sticking out of the membrane. Due to the slow washout / deactivation after LC-CoA application, it is sometimes impractical to wait until the currents fully return to the baseline, and in some cases, especially with TRPV6, there is substantial leftover current before the application of the next concentration. How much of this leftover current is due to lipid staying in the membrane due to its slow redistributing back to the aqueous phase where its concentration is zero between the applications, and how much is it due to the slow dissociation from the channel, is not known. Keeping these in mind, the 10 μM LC-CoA applied after 5 μM is technically not equivalent to applying 15 μM , but rather to 10 μM + the unknown concentration of the lipid in the membrane. Another complicating factor is that the dose-response for LC-CoA is extremely steep, basically no effect at 1 μM and presumably maximal effect at 10 μM . This is similar to what was observed earlier with K_{ATP} channels, see for example PMID: 12525701. Keeping all these in mind, a classical concentration response relationship is hard to establish with the accuracy of that obtained with water soluble compounds. We believe that the noted problems are inherent for experiments with lipids that act in the membrane, and the uncertainties in quantifying the effect LC-CoA do not significantly affect the conclusions of the paper.

Here the authors aim to show that PIP2 and LC CoA use the same binding site and mechanism to activate the channel. I think this is a fair hypothesis based on their data. However, I do not fully understand the approach.

The authors apply LC CoA to get a large TRPV5/TRPV6 current, then apply the soluble PIP2 analog at a concentration that produces a submaximal response to detect if the current increases.

My comments to this are:

1) diC8 PIP2 at the 25 μM concentration on its own produces a small current compared to the currents induced by LC CoAs. Could you expect to detect a potential current increase following PIP2 application? Especially given that we again do not reach steady state?

As explained earlier, reaching steady state is impractical, as the currents keep increasing during the application of LC-CoA. On the other hand, not having reached steady state means the channel is not yet maximally activated, so a further increase of activity is possible. With the fast acting DiC₈ PI(4,5)P₂ a further increase should be easy to detect even on a current that is increasing slowly. This is not what we see however, as DiC₈ PI(4,5)P₂ induces a small inhibition, which reached statistical significance with TRPV6 (0.013), but not with TRPV5 (p=0.115). The most likely explanation of this is that DiC₈ PI(4,5)P₂ is a partial agonist of these channels, which is well in line with the smaller effect of DiC₈ PI(4,5)P₂ compared to natural AAST PI(4,5)P₂ which has been documented earlier Velisetty et al, PMID 27291418.

The hypothesis of the paper, strongly supported by the structural work, is that LC-CoA and PI(4,5)P₂ act on the same binding site. The alternative hypothesis would be that they act on different sites, but in that case one may expect a potentiation, especially when we use submaximal LC-CoA, as it was shown for menthol and PI(4,5)P₂ on TRPM8 channels PMID:15852009, where subsequent structural studies confirmed that menthol and PI(4,5)P₂ act on different sites PMID:30733385. This is however not what our data with LC-CoA and DiC₈ PI(4,5)P₂ show.

2) Why do the authors choose a submaximal concentration of PIP2 for this experiment?

The dose response for DiC₈ PI(4,5)P₂ is relatively shallow, with an EC₅₀ of 79 μM (Velisetty et al 2016, PMID: 27291418), the currents do not saturate even at 200 μM, a concentration that makes the patches unstable. Rather than using a higher concentration of DiC₈ PI(4,5)P₂, we used lower concentrations of LC-CoA (Fig. 5 B,D). Given the fact that 25 μM DiC₈ PI(4,5)P₂ slightly inhibited currents after LC-CoA, it is quite unlikely that higher concentrations would have activated it. As explained in the previous point, the alternative hypothesis of acting at different sites would predict that LC-CoA potentiates the effect of DiC₈ PI(4,5)P₂, which is easier to detect at sub-maximal concentrations.

Given how similarly the two molecules appear to act on TRPV5 and TRPV6, can the authors offer any insights into why the length of the acyl chain does not seem to impact activation by PIP2 but is critical for acyl-CoA?

We did not discuss this in the manuscript, but our earlier work indicated that the length of the acyl chain also affects PI(4,5)P₂. Long acyl chain AAsT PI(4,5)P₂ at 10 μM induced currents well above cell attached levels, whereas 10 μM DiC₈ PI(4,5)P₂ induced much smaller currents, see for example Velisetty et al PMID 27291418. Also, the effect of AAsT PI(4,5)P₂ is essentially irreversible, whereas DiC₈ PI(4,5)P₂ is quickly reversible. Earlier work on Kir channels also showed dependence of the effect of PI(4,5)P₂ on acyl chain length, DiC₄ PI(4,5)P₂ having no effect, and DiC₈ PI(4,5)P₂ requiring higher concentrations than DiC₁₆ or AAsT PI(4,5)P₂ PMID: 10593888. We discussed this in the revised MS on page 8.

Standardize y axis in Fig 5 panels b-e.

We replotted the data using the same y axes.

In the methods, can the authors describe how the lipid stocks were prepared and handled?

We provided a detailed description of how the lipids were handled in the first paragraph of the methods.

I think the manuscript could do with a few supplementary items, like a figure describing

the cryo-EM data collection and processing, local resolution, density coverage in critical parts of the channel and figures showing additional traces (e.g., application of different acyl chain length CoAs in Fig1, TRPV6 traces in Fig5....)

We updated the revised manuscript with Supplemental information that includes representative traces (Fig. S1) and cryo-EM data processing (Fig. S2-3), as well as showing representative traces for both TRPV5 and TRPV6 in Figure 5.

I also found a few typos:

163 the TRPV5CoA Closed structure do not have such additional density / does

165 (S6 helix) (Fig. 4b), the same residues that was reported for the PI(4,5)P2-binding site / were

408 1:1:200:500. Lipids (soy polar lipids, Avanti) were dried under a nitrogen flow and store / stored

409 under vacuum until were resuspended in the detergent free wash buffer with the DMNG / they were

417 used to grid preparation. / used for

421 TRPV5 into nanodiscs sample was incubated / TRPV5 nanodisc sample

423 Scientific), 3 μ L of the samples were applied onto glow-discharged 1.2/1.3

Quantifoil / 3 μ L of the samples was

435 estimated max resolution below to 8 Å. / below 8 Å

436 of 200 micrographs and used as templated / template

440 Particles were oriented with TRPV5 apo map (25716) and refine with Non-uniform (NU) / refined

441 refinement in C4 symmetry (3.44 Å) to generate the first model, then was performed 3D / then 3D was performed

Will corrected those typos, thanks for noticing them

Reviewer #2 (Remarks to the Author):

LC-CoA esters are metabolic cofactors produced in certain cell and tissue types that have been shown to regulate a host of cellular processes, including modulation of ion channel activity. Here, the authors report TRPV5 and TRPV6 are activated by CoA species exhibiting long acyl chains and a 3' phosphate. The authors also solved the Cryo-EM structure of TRPV5 bound to an LC-CoA, which reveals that LC-CoA binds the same site as PI(4,5)P2 identified in their prior structures. When LC-CoA and the soluble PI(4,5)P2 analog DiC8 PIP2 are sequentially co-applied in excised patches, the authors do not observe a change in activation, which they interpret as LC-CoA occluding or preventing binding of DiC8 PIP2. Whole-cell patch experiments show that exogenous addition of LC-CoA is also able to maintain TRPV6 activity when PI(4,5)P2 is depleted from the membrane with VSP. The manuscript and its findings will be greatly elevated by addressing the following points:

Address physiological relevance: What is the evidence that the LC-CoAs used in this study are produced in the same cell types that endogenously express TRPV5/6 (currently there is a loose mention that TRPV5/6 may be in some pancreatic cells)? Are the LC-CoAs produced at a sufficient concentration in the plasma membrane (e.g., at levels comparable to those used in this manuscript) for this interaction to be physiologically relevant? If not, then the authors should shift their focus and make a compelling case for LC-CoA as a research tool and demonstrate its usefulness over PI(4,5)P₂. Either way, a cell line that is known to endogenously produce LC-CoAs would be ideal for the authors to use for a similar experiment to Figure 6. This would speak to whether endogenous levels of LC-CoAs are capable of gating TRPV5/6 since the channel would presumably retain activity if the LC-CoA is not affected by VSP. If this experiment is not feasible and/or it does not work, the authors should still address it.

We thank the reviewer for this suggestion. We are not aware of cell lines with high LC-CoA levels, but we agree with the reviewer showing that endogenously produced LC-CoA can modulate channel activity would make the manuscript stronger. Therefore, we adapted a previously published protocol (PMID: 16777940) to increase endogenous LC-CoA levels in HEK cells expressing TRPV6, and performed experiments similar to that in Figure 6. Our data show that increasing LC-CoA levels in HEK cells reduced inhibition by ciVSP, indicating that increased cellular LC-CoA levels can substitute for endogenous PI(4,5)P₂. The new data are shown in Figure 7 in the revised version, and we briefly discuss potential physiological relevance in the Discussion on page 8.

Contextualize and clarify existing results: The authors should better contextualize their structural data with their functional data. Currently, discussion of the lipid binding site is limited to residues conserved in PIP₂ binding, which predominantly engage the lipid headgroup. However, structural revelations about the importance of the acyl chain length or fatty acid saturation are not discussed. The authors should also discuss how their structure contributes to understanding the reversibility of LC-CoA binding as opposed to the nearly irreversible binding of PIP₂.

We extended the discussion on the acyl chain in the revised version (page 8). Briefly, for PI(4,5)P₂ most ion channels show a dependence on the length of the acyl chain, this has been documented on several ion channels. The importance of the acyl chains is most likely to increase the partitioning of the lipids to the membrane, where they exert their effects on the channels.

Additionally, the results in Figure 5 would benefit from a more critical analysis. At a saturating (10 μM) concentration of LC-CoA, the authors see a dip in TRPV5/6 activity when diC₈ PIP₂ is applied, which is significant for TRPV6. This could be explained by exchange of the better activating LC-CoA for the weaker activating DiC₈ PIP₂ and the authors should better interpret this dip.

The reviewer is correct that the dip is likely to be due to DiC₈ PI(4,5)P₂ being a somewhat less efficacious activator i.e. partial agonist, as we also explained in

response to reviewer one, and discussed it in the revised MS on page 8.

Given this, it is unclear why the authors observe no change in TRPV5/6 activity when a sub-saturating (3 μ M) concentration of LC-CoA is co-applied with DiC8 PIP2.

DiC8 PI(4,5)P₂ at 25 μ M is still a weaker stimulus than 3 μ M oleoyl-CoA, thus if they act on the same binding site, it is not expected to give a very strong additional activation. As a partial agonist, it is probably expected to exert less inhibition at submaximal stimulations this may be the reason why a clear decrease was not detected.

As also explained in the response to reviewer 1, the main reason this experiment was performed however is to test the alternative hypothesis that LC-CoA and PI(4,5)P₂ act on different binding sites, where the expectation would be potentiation, as described for example for menthol and PI(4,5)P₂ on TRPM8. The fact we did not observe potentiation is compatible with PI(4,5)P₂ and LC-CoA acting on the same binding site. See also the revised discussion section on page 8.

The authors should also address whether PIP2 instead of DiC8 PIP2 is a better reagent for this experiment - since PIP2 may have increased binding affinity and may be more comparable to LC-CoA in its activity.

AAST PI(4,5)P₂ acts slowly and essentially irreversibly, thus it would be impractical to use it for these experiments. We referred to this in the revised MS on page 3.

Finally, the authors should better explain how their data supports their conclusion that LC-CoA “occludes” PIP2. The authors propose both LC-CoA and PIP2 activate TRPV5/6 by binding the same site, thus they have the same effect on channel function, so perhaps “substitutes for” or “replaces” would be more accurate than “occludes”.

We removed the word occlude and used a different, more descriptive language in the revised manuscript.

Additional data/experiments that are needed to strengthen their findings:

1. The authors use DiC8 PIP2, a soluble PIP2 analogue, throughout their manuscript to make direct comparisons between PIP2 and LC-CoAs. The authors should directly compare TRPV5/6 activation by PIP2 to activation by diC8 PIP2 (and distinguish the diC8 PIP2 from endogenous PIP2 in their figures - current labeling is misleading). This will allow them to scale or calibrate their comparative results from LC-CoAs and diC8 PIP2. This scaling control is important because the authors appear to assume that the soluble short chain PIP2 analogue (diC8 PIP2) and PIP2 are equally capable of activating TRPV5/6, however, these lipids cannot be equal based on the authors' findings with LC-CoAs. LC-CoAs and PIP2 occupy the same site in TRPV5/6 and the authors show there is a strong effect of alkyl chain length on LC-CoA activation. DiC8 PIP2 has an alkyl chain length (C8) that was nonfunctional in an LC-CoA, which may explain why they needed a higher concentration of diC8 PIP2 than LC-CoA to activate

TRPV5/6.

As also mentioned to the response to reviewer 1, several of our earlier publications compared the effects of DiC₈ PI(4,5)P₂ and natural AAs PI(4,5)P₂ PMID: 21810903, 23300090 and 27291418. We discussed the acyl chain dependence of the effect of PI(4,5)P₂, on page 8, and relabeled all figures indicating that DiC₈ PI(4,5)P₂ was applied to avoid confusion.

2. This study would benefit from validation of the TRPV5 structures with mutagenesis to disrupt LC-CoA binding residues, followed by functional analysis. Can they make mutations that preclude LC-CoA binding, but not PIP2? While the authors were unable to solve the structure of TRPV6 bound to LC-CoA, the authors may be able to apply the same mutagenesis strategy to TRPV6 to strengthen their claim that TRPV6 uses a similar binding mechanism for LC-CoA as TRPV5.

Validating the LC-CoA binding site with mutations is problematic, as the LC-CoA binding site is essentially the same as the PI(4,5)P₂ binding site. Mutating residues in the binding site is likely to affect activation by both lipids. Given that the activity of both channels depend on either PI(4,5)P₂ or LC-CoA, a decrease in currents would be difficult to attribute to altering the effect of one, or the other.

In our initial studies, before we had the structure of TRPV5 with LC-CoA, we attempted to identify the LC-CoA binding site using the combination of computational docking and mutagenesis. Those results were very labor intensive, yet inconclusive, and we did not include them in the manuscript. Briefly, our docking results showed a very similar, yet not identical, location of LC-CoA in the structure of TRPV5 as PI(4,5)P₂ in the experimentally determined structures. Based on our docking, we found two positively charged residues that were in contact in our computational model with LC-CoA but not with PI(4,5)P₂. Mutating these residues in TRPV5 or TRPV6 however, had only marginal effects on LC-CoA activation, therefore we took a different route and attempted to determine the structure of LC-CoA with TRPV5. This attempt was successful, and explained the failure of the mutation efforts, as the two residues we mutated were not any closer to LC-CoA than to PI(4,5)P₂ in the experimentally determined structures, thus they were not expected to selectively decrease the effect of LC-CoA. Given the very high similarity of the PI(4,5)P₂ and LC-CoA binding site, validating the LC-CoA binding site is likely to be just as futile as our original efforts using mutagenesis.

3. To test the relative ability of LC-CoA and PIP2 each to displace the other, the authors should collect data where they activate with Lipid 1, coapply Lipids 1 and 2, then remove Lipid 1. The slope of the change in current from Lipid 1+2 to Lipid 2 would give the authors the displacement rate. These experiments should be performed with both LC-CoA and diC8 PIP2 as Lipid 1, allowing a more quantitative comparison of displacement rates to determine whether LC-CoA indeed has a higher affinity for the binding site and “occludes” PIP2, as the authors claim.

We think this experiment would be complicated to interpret for several reasons. First, the deactivation rate of the channels after activation by these lipids is likely to be a combination of the off rate of the lipid binding to the channel and the diffusion of the lipid out of the membrane. This is especially problematic with LC-CoA for which the deactivation is not only slow, but also highly variable. Second, DiC₈ PI(4,5)P₂ slightly inhibits, rather than activates after application of LC-CoA. Third, natural PI(4,5)P₂ activates these channels very slowly, and essentially irreversibly, thus the experiment with that lipid is also not practical.

Reviewers' Comments:

Reviewer #1:

Remarks to the Author:

The authors have answered my questions and addressed my concerns.

Reviewer #2:

Remarks to the Author:

The authors have provided a fair response to all raised comments. The new data in Figure 7 greatly increase the strength of the study and the updated language throughout improves clarity.